# LOVM: Language-Only Vision Model Selection

**Orr Zohar**
Stanford Univeristy
orrzohar@stanford.edu

**Shih-Cheng Huang**
Stanford Univeristy
mschuang@stanford.edu

**Kuan-Chieh Wang**
Stanford Univeristy
wangkua1@stanford.edu

**Serena Yeung**
Stanford Univeristy
syyeung@stanford.edu

## Abstract

Pre-trained multi-modal vision-language models (VLMs) are becoming increasingly popular due to their exceptional performance on downstream vision applications, particularly in the few- and zero-shot settings. However, selecting the best-performing VLM for some downstream applications is non-trivial, as it is dataset and task-dependent. Meanwhile, the exhaustive evaluation of all available VLMs on a novel application is not only time and computationally demanding but also necessitates the collection of a labeled dataset for evaluation. As the number of open-source VLM variants increases, there is a need for an efficient model selection strategy that does not require access to a curated evaluation dataset. This paper proposes a novel task and benchmark for efficiently evaluating VLMs' zero-shot performance on downstream applications without access to the downstream task dataset. Specifically, we introduce a new task LOVM: Language- Only Vision Model Selection, where methods are expected to perform both model selection and performance prediction based solely on a text description of the desired downstream application. We then introduced an extensive LOVM benchmark consisting of ground-truth evaluations of 35 pre-trained VLMs and 23 datasets, where methods are expected to rank the pre-trained VLMs and predict their zero-shot performance. Our code and dataset are available at https://github.com/orrzohar/LOVM

## 1 Introduction

Advancements in artificial intelligence (AI) have permeated diverse sectors, but applications in areas such as medicine or those with long-tail distributions often struggle to collect the sizable training datasets required for the standard supervised learning framework. Pre-trained vision-language models (VLMs) offer a promising solution, demonstrating robust performance on diverse downstream vision tasks without the necessity of large-scale labeled datasets [Radford et al., 2021, Jia et al., 2021]. However, the performance of VLMs can vary substantially across different tasks and domains, which undermines the reliance solely on benchmark dataset performance for effective VLM selection. Consequently, users aiming to *select a VLM* for custom downstream applications frequently face a predicament: the lack of established performance rankings for these specific, non-conventional tasks.

As the number of pre-trained VLMs increases (see Fig. 1 [Ilharco et al., 2021] ), the challenge of model selection escalates. Exhaustive evaluation of all available VLMs on a novel application requires first the collection of a labeled dataset for evaluation, and is also time and computationally demanding. However, many users lack the resources or technical proficiency to collect and label an evaluation dataset and subsequently evaluate all available VLMs. Consequently, the development of methods that efficiently select the most suitable model for a given task without relying on access to the downstream task dataset has become critically important.

37th Conference on Neural Information Processing Systems (NeurIPS 2023) Track on Datasets and Benchmarks.

Recent studies have demonstrated that text embeddings from VLMs can be used as a proxy for their corresponding image embeddings in various downstream tasks, including classification and error slice discovery [Zhang et al., 2023, Eyuboglu et al., 2022, Jain et al., 2022]. Specifically, although Liang et al. [2022] has shown that there exists a modality gap between text and image embeddings generated from VLMs, the geometry of this modality gap permits cross-modality transferability. This phenomenon allows text to serve as a proxy to corresponding images and vice versa. Therefore we aim to explore the utilization of cross-modality transferability to estimate VLM performance on a novel vision task using text alone.

Herein, we propose a novel problem setting - Language-Only VLM selection (LOVM) as a novel model selection task. In the LOVM task, methods are expected to select the optimal VLM and predict its expected performance given only a text description of a downstream vision task/application, (see Fig. 2). **Importantly, LOVM eliminate the need to gather, organize, and annotate custom datasets**, thereby greatly simplifying the model selection process for downstream users. Under the LOVM paradigm, machine learning practitioners could select the optimal VLM and deploy it in a novel application without ever collecting and annotating a custom dataset. To facilitate the development of LOVM methods in the future, we collected a large dataset of ground-truth evaluations of 35 pre-trained VLMs on 23 datasets. We then introduce the appropriate evaluation protocol and method quality metrics to allow the evaluation and comparison of future LOVM methods.

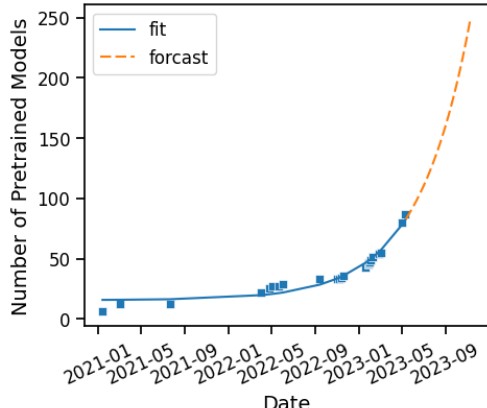

Figure 1: **LOVM Motivation.** Number of pretrained VLMs released on open-clip over time.

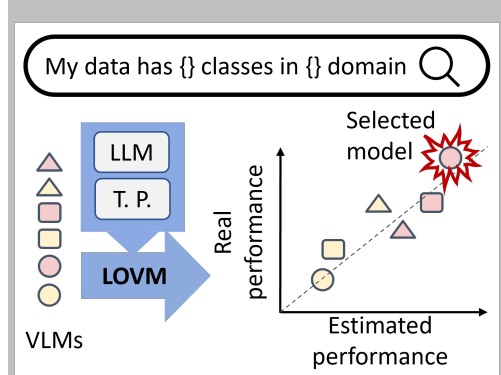

Figure 2: **An overview of an application for LOVM methods.** A user can type into a search bar the details of the desired task, and LOVM methods evaluate and rank the available models.

To show that such a challenging task is possible, we provide simple baselines that utilize readily available large language models to generate 'text datasets' for a given vision task. By utilizing the cross-modality transferability phenomenon, we show how simple baselines can be derived by utilizing the cross-modality transferability phenomenon. Our results show that text prompting may be an effective means of estimating zero-shot performance, showing that such a challenging task is possible while providing a baseline for future research.

The contributions of this study can be summarized as follows:

- We propose a novel problem setting, **LOVM**: Language-Only VLM selection and performance prediction. LOVM methods are expected to perform both model selection and performance prediction using only a text description of the desired zero-shot application.

- We provide a benchmark consisting of 35 pre-trained VLMs and 23 datasets. We evaluated all dataset-VLM combinations and reported their corresponding performance, which is used as the ground truth when training and evaluating LOVM methods. We also introduce the corresponding evaluation metrics and protocols.

- In developing the LOVM baselines, we introduce several novel methodological contributions, such as using LLM models to generate text proxies for images. Our text-based methods outperform simple baselines - such as ImageNet benchmarking, showing the promise of the direction of LOVM.

- By analyzing text-based score trends, we draw insights into VLM behavior and shed light on why ResNet-based models perform better on datasets with low visual diversity.

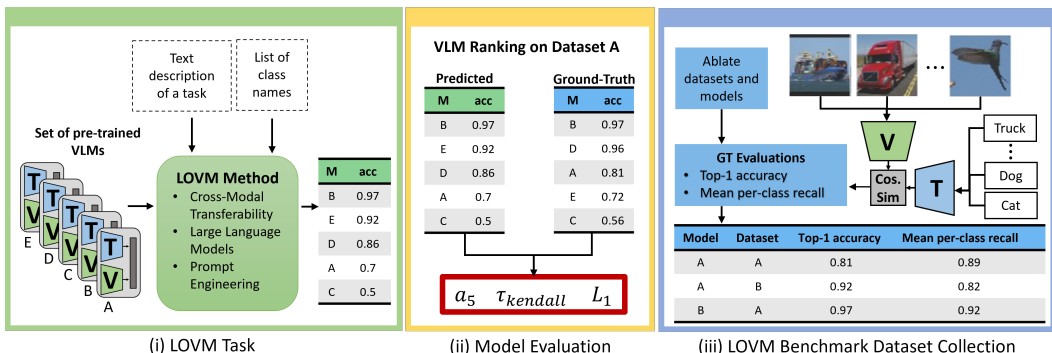

Figure 3: **Language-Only Vision Model Selection Overview.** (i) **Task.** a LOVM method is given a set of pre-trained VLMs, a text description of the desired task, and the list of the classes of interest. Given these, LOVM methods are expected to **rank** and **predict the performance** of all the available models on the downstream task. (ii) **Evaluation.** Given the *predicted (green)* and *ground-truth (blue)* VLM ranking and performance, we evaluate the LOVM method's performance by accepted list ranking and accuracy metrics. (iii) **Data Collection.** We exhaustively evaluated the selected 35 VLMs on the selected 23 datasets to produce the ground-truth (image-based) evaluations.

## 2 Language-Only Vision Model Selection

In order to train and evaluate LOVM methods, we need the ground-truth (GT) zero-shot performance, i.e., image-based evaluation of many VLMs (differing by architecture and pre-training) on many tasks and datasets. Once collected, we can develop and evaluate LOVM methods. An ideal LOVM method should be able to select the best performing VLM for a downstream vision task and estimate the performance directly from text embeddings, eliminating the cost of image-based model selection. The VLM, dataset selection criterion, and dataset collection procedure are detailed in Sec. 2.1. Finally, the evaluation protocol of LOVM methods is described in Sec. 2.2. For a discussion on why we only evaluate zero-shot performace, see App. Sec. D.

**Background.** We first recap how VLMs are used as in zero-shot vision tasks. Given a pre-trained VLM $v$, along with an image $X \in \mathcal{X}$ or text $Y \in \mathcal{Y}$ input, we can obtain their $L_2$-normalized embeddings $\boldsymbol{x}$ or $\boldsymbol{y}$ from the image encoder $f_x : \mathcal{X} \mapsto \mathbb{R}^n$ or the text encoder $f_y : \mathcal{Y} \mapsto \mathbb{R}^n$, where $n$ is the dimension of the shared multi-modal embedding space. To use a model $v$ on a particular task, one encodes the class prompts, $Y^c$ for class $c$ using the model's text encoder, producing the class embeddings $\boldsymbol{y}^c = f_y(Y^c)$. To produce the final class prediction, one calculates the cosine similarity of an image embedding with all the corresponding text embeddings to predict the class logits.

**Task Definition** In the LOVM task, for any downstream application/dataset $d$, methods are given a set of pre-trained VLMs, $\boldsymbol{V} = \{v_0, v_1, ..\} \in \mathcal{V}$, a text description of the downstream task $Y_d$ (e.g., classification) and a list of the desired class names $Y_d^c, \forall c \in C_d$ where $C_d$ is the number of classes in task $d$. Given this, LOVM methods are expected to **rank** and **predict the accuracy** of the set of models (see Fig. 3, i):

$$p_{v,d} = f_{\text{LOVM}}(v, \{Y_d^c\}_{c=1}^{C_d}, Y_d), \ \ \forall \, v \in \boldsymbol{V}, \tag{1}$$

where $p_{v,d} \in \mathbb{R}$ is the relative/absolute performance of model $v$ on dataset $d$.

### 2.1 Data Collection and Benchmark Construction

To train and evaluate LOVM methods, we need the **zero-shot** ground-truth performance of many VLM models on many downstream datasets. We, therefore, selected 35 VLMs and 23 Datasets and then performed image-based evaluations of each model on all the datasets - a total of 805 evaluations using the same prompting strategies discussed by Radford et al. [2021], See Fig. 3, iii. **These ground truth zero-shot image-based model rankings and accuracies constitute the bulk of our benchmark.** The proposed LOVM benchmark consists of the aforementioned evaluation tables as well as the per-dataset prompting templates, class names, and domain descriptions.

**Selected Datasets.** The proposed LOVM benchmark utilizes a heterogeneous assortment of 23 datasets. These datasets exhibit variability in the number of classes, their target tasks, and corresponding domains. The benchmark encompasses a comprehensive range of tasks such as classification, scene understanding, geolocalization, and object counting, rendering it extensively applicable across many applications. Further, the datasets span diverse domains, including natural, satellite, text, and medical images (See Tab. 1). To ensure maximal compatibility, we have opted for tasks that permit the utilization of the same VLM architecture, precluding any requisite alterations or additional training. This approach necessitated the exclusion of tasks such as segmentation and object detection, which mandate additional training modules, introducing extraneous noise during the evaluation of VLM performance.

Table 1: Details on the different datasets used, including the number of classes, tasks, and domain.

| Dataset | Classes | Task | Domain |
|---|---|---|---|
| Imagenet | 1000 | classification | natural image |
| Stanford Cars | 196 | classification | natural image |
| Flowers102 | 102 | classification | natural image |
| CIFAR100 | 100 | classification | natural image |
| GTSRB | 43 | classification | natural image |
| VOC2007 | 20 | classification | natural image |
| Oxford Pets | 37 | classification | natural image |
| STL10 | 10 | classification | natural image |
| DTD | 46 | classification | textural image |
| RESISC4 | 45 | classification | satellite images |
| EuroSAT | 10 | classification | satellite images |
| MNIST | 10 | classification | hand-writing |
| Retinopathy | 5 | classification | retina scan |
| PCam | 2 | classification | histopathology |
| SUN397 | 397 | scene und. | natural image |
| Country211 | 211 | geolocation | natural image |
| SVHN | 10 | OCR | natural image |
| CLEVR-C | 8 | object counting | natural image |
| CLEVR-D | 8 | distance est. | natural image |
| DMLab | 6 | distance est. | synthetic |
| FER2013 | 7 | fac. exp. rec. | natural image |
| KITTI | 4 | distance est. | natural image |
| Rendered SST2 | 2 | OCR | text image |

**VLM Candidates.** We utilize the open-clip library [Ilharco et al., 2021], a diverse collection of pre-trained VLMs spanning various architectures, including but not limited to CLIP and CoCa models, and utilizing encoders such as ResNet, ConvNext, and ViT. These models have undergone pre-training on various datasets, such as WIT [Radford et al., 2021], LAION 400m, and LAION 2b [Schuhmann et al., 2022], with different hyperparameters. From the 87 models currently available, we have carefully selected 35 for our study. A comprehensive list of all models used in this benchmark can be found in the App. Tab. 4. We avoided incorporating additional multi-modal models, such as BEIT[Wang et al., 2023] and VLMO [Bao et al., 2022], as these models utilize a shared text-image encoder and, therefore, cannot be evaluated on the same datasets as CoCa and CLIP. Utilizing models from the open-clip library ensures maximum compatibility and reproducibility in our work. Currently, CLIP models comprise a significant portion of VLMs employed in practice.

## 2.2 LOVM Evaluation Protocol

On our benchmark, methods are expected to rank 35 pre-trained multi-modal models that differ in architecture and pre-training datasets on 23 target datasets, and compare these rankings to the ground-truth rankings (see Fig. 3 (ii)) and report the performance for each of the 23 datasets as well as their averaged values.

**Model Ranking.** When evaluating model ranking on a particular dataset, one has access to the performance of all the models on all the datasets besides the one being evaluated. We use the following metrics:

- *Top-5 Recall* ($R_5$) – We used $R_5$ to evaluate a LOVM method's model ranking capability. It is defined as the ratio of correctly identified models.

- *Kendall's Rank Correlation* ($\tau$) – We used $\tau$ to evaluate a LOVM method's model selection capability and give s fine-grained picture of how well the method ranked the high-performing models and is defined as Kendall's rank over the top-5 selected models.

**Performance Prediction.** When evaluating a model's prediction on a dataset, the GT performance of that model on all datasets and the performance of all models on that dataset are held out.

- *Mean Absolute Error* ($L_1$) – We used $L_1$ to evaluate a LOVM method's performance prediction capability. Specifically, we compute the $L_1$ loss of all models' predicted vs. actual mean per-class recall/top-1 accuracy.

## 3 LOVM Baselines

The assessment of model performance in traditional supervised methods often relies on benchmark dataset performance. Given that most pre-trained vision-language models (VLMs) are evaluated on ImageNet, it is convenient to utilize it as a baseline for comparison (This is our ImageNet Benchmark baseline). Alternatively, a large language model could generate many probable image captions, which could be encoded using the different VLMs text encoder, producing the corresponding text embeddings. Treating these embeddings as image-proxies, one can calculate different widely-accepted scores (see Sec. 3.2) and fit a linear regression model to predict performance or rank VLMs. Specifically, from every VLM-dataset combination, one extracts these scores and then fits the model:

$$p_{v,d} = \boldsymbol{w} \cdot \boldsymbol{s}_{v,d} + b, \tag{2}$$

$$s_{v,d}^i = f_{\text{feat}}^i(v, \texttt{TextGen}(\{Y_d^c\}_{c=1}^{C_d}, Y_d)), \tag{3}$$

where $p_{v,d} \in \mathbb{R}$ is the relative/absolute performance of model $v$ on dataset $d$, $\boldsymbol{w}, b$ are the weights and bias of the linear model. $s_{v,d}^i$ is the $i$-th element in the score vector, $\boldsymbol{s}_{v,t} = [s_{v,d}^1, s_{v,d}^2, ...]^T$, produced by the corresponding feature/score function $f_{\text{feat}}^i$. The function $\texttt{TextGen}$ is a function that generates text given the class names, $\{Y_d^c\}_{c=1}^{C_d}$ and task description $Y_d$ of the desired task/dataset $d$.

We discuss the different scores, $s_{v,d}^i$, in Sec. 3.2 and the $\texttt{TextGen}$ function in Sec. 3.1. To evaluate model rankings on a dataset, we hold out the data for that particular dataset and fit a linear model on all the other datasets. Meanwhile, to evaluate the performance prediction of some model on a particular dataset, we hold out the data for that dataset and model and fit a linear model on the remaining combinations. We refer to the baselines by the combination of scores used in the model.

### 3.1 Text Data Generation

The impressive progress in large language models (LLMs) [OpenAI, 2023, Touvron et al., 2023] has rendered the generation of potential - and realistic - 'image captions' practically limitless, thus rendering text data generation remarkably attainable. In our study, we employ GPT-3.5, tasked to produce two distinct text-based datasets, each corresponding to a given vision task. These generated datasets serve as the foundation for extracting essential features for our task.

**Captions Dataset.** To generate the captions dataset, $\boldsymbol{D}^{\text{cap}}$, we prompt an LLM to generate realistic - but confusing - captions for images containing the user-provided classes in the user-provided domain. We extracted the dataset description and class names from each dataset and prompted the LLM:

> Generate long and confusing image captions for the {domain} domain, which will be used to evaluate a Vision-Language Model's {task} performance.
> Generate 50 long, domain-specific captions for {classname}:

Where we assume the user supplies the target domain and task description. For examples of different dataset's domain and task, see Tab. 1.

**Synonyms Dataset.** Prior studies have already leveraged synonyms to evaluate LLMs [van der Lee et al., 2023]. For example, if an VLM has seen many instances of the class 'chair' referenced as a 'chair', 'seat', etc., we expect these embeddings to be closely located in the shared embedding space. To evaluate this aspect of the VLM using text, we prompt an LLM to generate a list of semantically similar/synonyms for every object class:

> Please list the superclasses/synonyms for {classname}. For example:
> chair: [furniture, seat, bench, armchair, sofa]
> {classname}:

We collect the results from this prompt to form the synonyms dataset, $\boldsymbol{D}^{\text{syn}}$.

### 3.2 Text-Derived Scores

There are many widely reported metrics for model transferability, dataset difficulty, and dataset granularity scores developed on image embeddings. We extract different commonly used features/metrics from the text dataset embeddings and calculate their text-only counterparts.

Table 2: **LOVM Benchmark**. We evaluate our method's performance over 23 datasets and 35 pre-trained models, and when predicting the top-1 accuracy and mean per-class recall (averaged over all datasets, for the per-dataset breakdown, see App. Tab. 5 and 6 ). INB - ImageNet Baseline, C - Text Classification scores, G - Granularity scores. As can be seen, mixed approaches achieve the best VLM ranking and performance prediction.

| used scores | mean per-class recall | | | top-1 accuracy | | |
|---|---|---|---|---|---|---|
| | $R_5(\uparrow)$ | $\tau(\uparrow)$ | $L_1(\downarrow)$ | $R_5(\uparrow)$ | $\tau(\uparrow)$ | $L_1(\downarrow)$ |
| INB | 0.504 | 0.186 | 0.228 | 0.452 | 0.177 | 0.220 |
| C | 0.252 | 0.058 | 0.182 | 0.226 | 0.058 | 0.176 |
| G | 0.270 | -0.014 | **0.141** | 0.252 | -0.014 | 0.144 |
| G+C | 0.270 | -0.014 | **0.141** | 0.252 | -0.014 | 0.144 |
| INB+C | 0.513 | **0.200** | 0.182 | 0.452 | **0.223** | 0.176 |
| INB+G | **0.548** | 0.197 | **0.141** | **0.461** | 0.096 | **0.140** |
| INB+G+C | **0.548** | 0.197 | **0.141** | **0.461** | 0.096 | **0.140** |

**Text Classification Scores (C).** We use the generated captions dataset as image proxies and evaluate the resulting model performance. Specifically, we replace the images with the generated image captions and evaluate each model's text top-1 accuracy (**text-acc1**) and f1-score (**text-f1**).

**Dataset Granularity Scores (G).** Cui et al. [2019] introduced the use of two widely used dataset granularity measures for image classification, Fisher criterion [Fisher, 1936], $\phi_{\text{fisher}}$ and Silhouette score [Rousseeuw, 1987], $\varphi_{\text{sil}}$, and their normalization constant, Class Dispersion score, $\rho_{\text{disp}}$. The Fisher criterion measures the degree of similarity of the classes or the extent of their separation. The Silhouette score is a well-established metric used to quantify the tightness of the same-class samples to the separation of different-class samples. The Class Dispersion score quantifies the degree of same-class tightness or data cone radius.

Recently, van der Lee et al. [2023] has shown that synonym consistency can be used in large language models to correlate the degree of familiarity of the model with a particular concept. Using the Synonym dataset, we compare the cosine similarity between the text embedding of each class and its corresponding synonyms. A high Synonym Consistency score, $\gamma_{\text{syn}}$, between the class and its corresponding synonyms indicates that the model is aware of the semantic meaning of the class.

**ImageNet Benchmark (INB).** We use the Imagenet performance of a VLM as the simplest baseline for our LOVM methods. Here we assume that the performance of each model on all the downstream tasks is exactly equal to the ImageNet performance. Methods often report ImageNet zero-shot classification performance and it is therefore reasonable to believe we have this.

For detailed definitions of these metrics, see App. Sec. B.

## 4 Experiments and Results

In Sec. 4.1, we evaluate the model selection capabilities of the proposed baselines on the LOVM benchmark. In Sec. 4.2, we evaluate the proposed baselines' performance prediction capabilities. We then analyze score trends and draw insights in Sec. 4.3.

### 4.1 Model Selection

A core aspect of this benchmark is model selection, as it allows the user to quickly and easily select the optimal model for the desired downstream task. From Tab. 2, we can see that, when predicting/ranking by the models mean per-class recall, the (C+G)-baseline can achieve a top-5 recall of 0.270, indicating that, on average, more than one model is correctly ranked as a top-5 performing model. Meanwhile, the INB-baseline had a $R_5$ of 0.504. Combining the text and ImageNet scores, the (INB+G)-baseline achieves the highest recall of 0.548, a $\sim 15\%$ improvement over the INB-baseline. To observe more fine-grained ranking capability, studying Kendall's rank correlation, the (G+C)-, INB-, and (INB+C)-baselines achieve a $\tau$ of $-0.014, 0.186$, and $0.200$, respectively.

Similar results can be seen when predicting the top-1 accuracy. The consistent improvement of the baselines over the INB-baseline indicates the utility of both text-based and benchmark features. Interestingly, C-score (or the text-acc1) appears to be more influential in predicting/ranking model's the top-1 accuracy than the mean per-class recall. To show that changing the LLM does not affect these results, we re-ran our experiments with a different LLM and report the results in Sup. Sec. C.3

### 4.2 Performance Prediction

Based on Tab. 2, it is clear that the granularity scores (G) are instrumental to predicting a model's top-1 accuracy and mean per-class recall. The G-baseline approach can achieve an average $L_1$ error of 0.145 and 0.141 for predicting the mean-per-class recall and top-1 accuracy, respectively. Adding any other scores does not lead to an improvement in performance prediction. The INB-baseline, which uses Imagenet performance as prediction, leads to a much higher $L_1$ error of 0.228 and 0.220 compared to the text-base baselines (text-based performance estimation outperformed INB-baselines by $\sim 36\%$). Finally, adding the

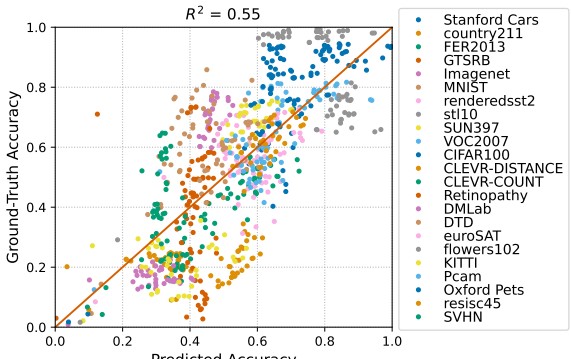

Figure 4: **Predicted vs. Ground-Truth Accuracy**. Predicted vs. actual top-1 accuracy on the proposed LOVM benchmark.

ImageNet benchmark score to the text features in the Unified baseline did not improve the $L_1$ compared to the text-only baseline. This is expected as the imagenet performance cannot be used to predict the performance on a different dataset. Fig. 4 shows the predicted vs. ground-truth accuracy. Our approach had a $R^2$ score (or coefficient of determination) of 0.55, showing significant room for improvement in accuracy prediction. To show that changing the LLM does not affect these results, we re-ran our experiments with a different LLM and report the results in Sup. Sec. C.3

### 4.3 Insights into VLM Behavior

In this section, we visualize the dependence of the text-derived features on the pre-training datasets and model architectures while averaging them across the different datasets (see Fig. 5).

**Model Size.** From studying Fig. 5, we can we can identify a clear trend of Fisher criterion and Silhouette score improving with model size, while Class Dispersion score and Synonym Consistency score degrade with model size. Silhouette score quantifies the degree of inter-class overlap or the degree of overlap between different classes in the embedding space. As the model size of the visual encoder increases, the embeddings from different classes become more and more orthogonal, decreasing the inter-class overlap. Fisher criterion quantifies the degree of granularity a model perceives the target datasets to be. As model size decreases, Fisher criterion decreases, or the degree of perceived granularity increases. Class Dispersion score quantifies the degree of intra-class dispersion, or how similar embeddings of the same class are. Specifically, as we increase model size, Class Dispersion score decreases, and therefore the class embeddings become more varied, effectively expanding the class cone radius. Synonym Consistency score quantified the closeness of a class to its synonyms and behaved similarly to Fisher criterion.

**Pre-training Dataset.** When studying the effect of pre-training dataset size, it is clear that there is a positive correlation between pre-training dataset size and all of the metrics when comparing models of the same size. As the pre-training dataset increases, the intra-class simularity increases more rapidly than the inter-class simularity, hence effectively different classes are more seperated. Specifically, Fisher criterion and Silhouette score increase, or the degree of perceived granularity decreases, and embeddings from different classes become less orthogonal, increasing the inter-class overlap. As the pre-training dataset size increases, Class Dispersion score increases and the intra-class dispersion is more condensed, leading to a smaller effective radius of a class dataset cone. Interestingly, larger models are more affected by the increase in dataset size (as seen by the large slope of ViT-L compared to ViT-B) - which could explain previous works' observation that larger models benefit more when trained on larger datasets [Fang et al., 2022].

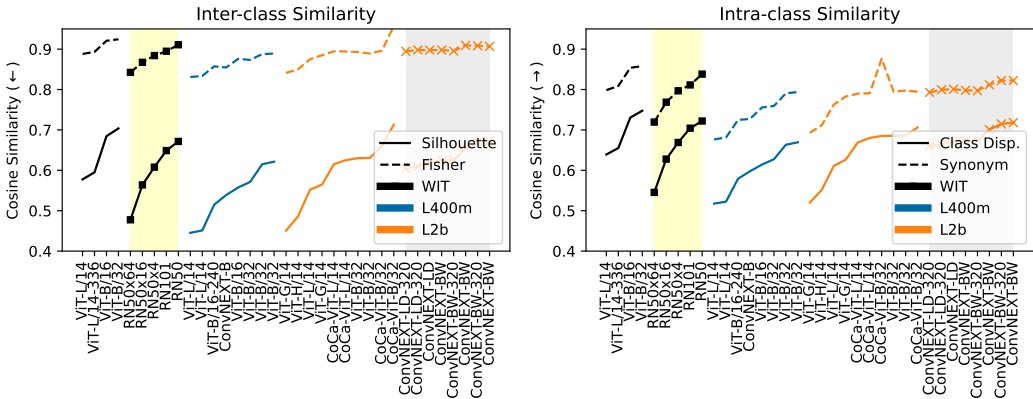

Figure 5: **Analyzing Score Trends.** Average text scores dependence on pre-training datasets and model architecture on our text-derived scores. (left) scores quantifying inter-class similarity (right) scores quantifying intra-class similarity. ResNet ( ▪ ) and ConvNext ( ✕ ) based models are grouped separately to evaluate their effect on the score trends.

**Model Architecture.** Pre-training datasets and model architectures significantly influence each other. ResNets and ViTs, for instance, consistently demonstrated differing behaviors and appeared to reside at distinct points on the class separation-dispersion trade-off curve. In particular, ResNets displayed lower Class Dispersion score and Silhouette score, indicating challenges in encoding instances of the same class within the feature space compared to ViTs. This may account for ResNets' superior performance on datasets with low visual variation, like MNIST; as the visual variation is relatively low, we would not expect the Class Dispersion score to be the limiting factor in model performance, making them less affected by this aspect of the dataset. Intriguingly, ConvNEXT models exhibited characteristics more in line with ViT-base models than ResNet-based ones. What leads to variation between WIT and L400m remains unclear, necessitating further investigation.

## 5 Related Work

**Vision-Language Models.** The field of vision-language models (VLMs) has witnessed significant progress in recent years, particularly with the introduction of contrastive pre-trained VLMs such as CLIP [Radford et al., 2021]. These models leverage large-scale datasets of aligned image-caption pairs to obtain shared embedding spaces that capture rich visual and textual features. The learned image and text encoders from these VLMs have demonstrated impressive feature extraction capabilities and even set state-of-the-art zero-shot performances. However, the performance of VLMs can vary significantly across different datasets, especially when there exists a domain, content, or distribution shift [Fang et al., 2022]. As the number of model architectures & pre-training datasets [Ilharco et al., 2021, Schuhmann et al., 2022] increase, it is challenging to select a pre-trained VLM, as good performance on existing benchmarks does not always translate to the downstream task. Therefore, there is a need to develop strategies that can estimate VLM performance on a new task without requiring an exhaustive evaluation of these models using the target dataset.

**The Cross-Modality Transferability Phenomenon: Text as a Proxy For Images.** While these VLMs aim to project representations from different modalities into a shared embedding space, Liang et al. [2022] found that corresponding image and text pairs don't completely overlap in the embedding space. Instead, a "modality gap" exists between the image embeddings and text embeddings subspace. Subsequently, Zhang et al. [2023] has found that this gap can be approximated as an orthogonal constant between true pairs of image and text and is, therefore, parallel to the decision boundaries for a given modality. This suggests that cross-modality transferability - using one modality as input to the other's classifier - is possible for these contrastively pre-trained VLMs. Several studies have demonstrated the utility of the cross-modality transferability phenomenon in different tasks. For instance, Domino leveraged the cross-modal embeddings to identify error slices and generate natural language descriptions of the error slices [Eyuboglu et al., 2022]. Similarly, Jain et al. [2022]

used these embeddings to discover model failure directions in the multi-modal embedding space. Meanwhile, Zhang et al. [2023] proposed the DrML, which diagnoses and rectifies vision classifiers using natural language inputs. In this study, we also use text as a proxy for images, but for the novel task of ranking and estimating VLM performance.

**Unsupervised Model Selection.**  Unsupervised model selection was recently introduced by Sun et al. [2021], to select the best model for a new target domain without utilizing labeled data. Their work only considered domain (and not content) shifts and proposed constructing a proxy dataset that captures/closely approximates this shift. This proxy dataset is constructed by minimizing different dataset statistics using several labeled datasets. Evaluating models on this proxy set performs well for model selection/ranking. However, such a strategy is limiting in the setting of evaluating VLMs - the size of these models and their pre-training datasets makes it too computationally expensive to achieve the desired goal of evaluating model performance on *any* downstream task.

**Unsupervised Accuracy Estimation.**  Unsupervised or label-free accuracy estimation aims to estimate classifier model performance with only access to the unlabeled test set of a new task. Platanios et al. [2017, 2016] proposed strategies to apply probabilistic modeling approaches, such as the probabilistic logic or Bayesian modeling, to analyze and aggregate predictions from multiple classifiers. Other works approach this task by fitting models on feature statistics of the target dataset [Risser-Maroix and Chamand, 2023]. Some studies evaluated model agreement, where many classifiers are used on the target dataset, and the degree of agreement was correlated with model performance [Chen et al., 2021, Jiang et al., 2022] Other approaches for unsupervised accuracy estimation include training a neural network on the weight distribution statistics [Unterthiner et al., 2020] or composing a meta-dataset with available datasets, such that the meta-dataset matched some target dataset statistics [Deng and Zheng, 2021]. Some have attempted to craft embedding-based scores, trying to quantify the separability of clusters in the embeddings spaces [Pándy et al., 2022, Ding et al., 2022]. All these methods assume access to the unlabeled dataset of the target task. Instead, our method only requires text descriptions of the novel task to estimate the model's performance.

# 6   Conclusion

In this work, we introduce a new problem setting and task LOVM, which aims to select the best-performing VLMs for a downstream vision task by only using its textual description. To demonstrate the feasibility of such a task, we show how large language models, in combination with the cross-modal transferability phenomenon, can be leveraged for such a task. We exhaustively test these methods on the proposed LOVM benchmark, consisting of 35 VLMs and 23 benchmark datasets. Our findings validate the viability of our proposed LOVM task, with unified (both text scores and ImageNet benchmarking) baselines outperforming the ImageNet benchmarking baseline. This suggests that text-based model selection methods (i.e., LOVM methods) provide additional benefits to baseline selection based on a model's performance on ImageNet. Furthermore, we found that the granularity-based scores influence performance prediction and modal ranking more greatly. These findings bolster the research direction of developing methods for VLM selections using only text.

Our proposed LOVM benchmark aims to foster this research direction. We see two promising avenues for future research: (i) improving text-based classification correlation with ground-truth accuracy by either text generation, evaluation metrics, or cross-modal transferability, and (ii) introducing new granularity and transferability scores to the text-only paradigm. Namely, we anticipate the development of methods improving over our proposed baselines presented in Tab. 2. Our work aims to facilitate future research in this area and provide a more accurate and reliable means of comparing pre-trained VLMs, accelerating their utilization in downstream applications.

For a discussion about broader and potential negative societal impacts please see App. Sec E.

**Acknowledgments.**  We gratefully acknowledge the computational credits provided by Google Cloud Platform through Stanford's HAI Institute for Human-Centered Artificial Intelligence. We also thank the Knight-Hennessy Scholars Foundation for generously funding Orr Zohar.

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
