# LOVM: Language-Only Vision Model Selection
# Appendix

## Table of Contents

## A  LOVM Benchmark Details

We evaluated 35 on 23, a total of 805 evaluations. This constituted the bulk of our compute with a total of 4 days on an nvidia V100 instance. Evaluations were carried out using the CLIP_benchmark repository (`https://github.com/LAION-AI/CLIP_benchmark`).

### A.1  LOVM Benchmark - Datasets

The proposed LOVM benchmark comprises of 23 datasets, which were selected to maximize diversity. Specifically, these datasets vary in the number of classes (2 to 1000), their target tasks, and domains. The benchmark encompasses a comprehensive range of tasks such as classification, scene understanding, geolocalization, object counting, and more, with the goal of rendering it extensively applicable across many applications. Further, the datasets span diverse domains, including natural, satellite, text, and medical images (See Tab. 3 for a comprehensive account of the datasets and their source). To ensure maximal compatibility, we have opted for tasks that permit the utilization of the same VLM architecture, precluding any requisite alterations or additional training. This approach necessitated the exclusion of tasks such as segmentation and object detection, which mandate additional training modules, introducing extraneous noise while evaluating VLM performance. However, it is worth noting that previous transferability works have shown that these approaches may generalize to more complex applications such as semantic segmentation [Pándy et al., 2022, Agostinelli et al., 2022].

### A.2  LOVM Benchmark - Vision-Language Models

Tab. 4 presents a list of models and their corresponding pre-training datasets used in the LOVM benchmark. We utilize the open-clip library [Ilharco et al., 2021], a diverse collection of pre-trained VLMs spanning various architectures, including but not limited to CLIP and CoCa models, and utilizing encoders such as ResNet, ConvNext, and ViT. These models have undergone pre-training on various datasets, such as WIT [Radford et al., 2021], LAION 400m, and LAION 2b [Schuhmann et al.,

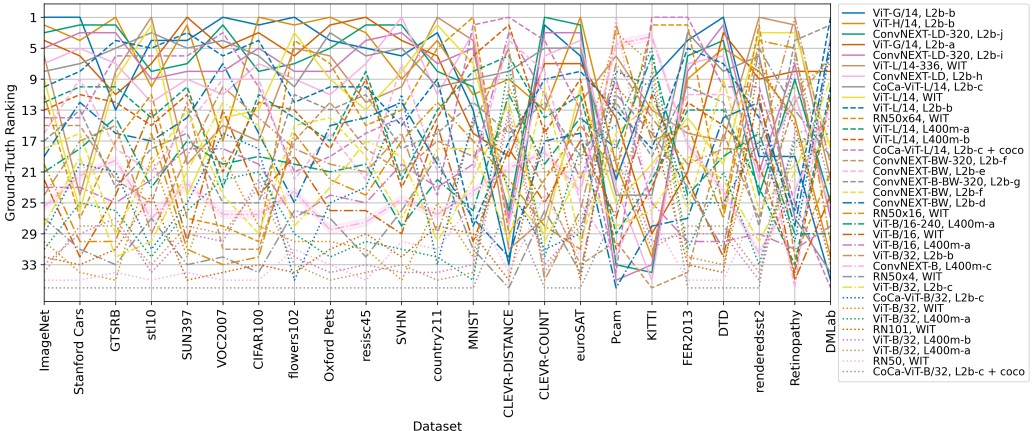

Figure 6: **Ground-Truth VLM Ranking.** As can be seen, there is a lot of variation in the ground-truth model ranking across both the natural image (left) and other (right) benchmarks.

2022], with different hyperparameters. From the 87 models currently available, we have carefully selected 35 for our study. A comprehensive list of all models used in this benchmark can be found in Tab. 4. We avoided incorporating additional multi-modal models, such as BEIT[Wang et al., 2023] and VLMO [Bao et al., 2022], as these models utilize a shared text-image encoder and, therefore, cannot be evaluated on the same datasets as CoCa and CLIP. Using models from the open-clip library ensures maximum compatibility and reproducibility in our work. Currently, CLIP models comprise a significant portion of VLMs employed in practice. Tab. 4 includes 35 entries, each identified by an ID number. The first four columns indicate the ID number, model name, model abbreviation, and pre-training dataset name. The fifth column shows the abbreviation of the pre-training dataset name. The models listed in the table include ResNet (RN50, RN101, etc.) and Vision Transformer (ViT) with different sizes (B/32, B/16, L/14, etc.), and the pre-training datasets include OpenAI's WIT dataset and two variants of LAION (L400m and L2b) datasets.

### A.3 LOVM Benchmark - Ground-Truth Model Ranking

To evaluate the validity and generalizability of the LOVM benchmark, we first present the ground-truth model ranking over all datasets to show that the model order is not constant across the datasets. We organized the benchmarks from natural image classification (Fig. 6, left) to non-natural image / non-classification benchmarks (Fig. 6, right). As depicted in Fig. 6, the distribution exhibits a non-uniform pattern, indicating the utility of LOVM methods and the importance of VLM selection methods in general. Interestingly, ranking variations are more significant on the non-natural image / non-classification benchmarks. This exemplifies the need for LOVM methods to contend with content shift (i.e., changing what classes are in the target domain) and domain/task shift.

## B   Baseline Details

Fig. 7 shows an overview of our baselines. We first describe the prompting protocol used in Sec B.1. We then give an in-detail description of the scores used in the study in Sec. B.2. In Sec. B.3, we give additional details for the text dataset generation. Finally, in Sec. B.4, we describe how we use noise to corrupt the text caption dataset when calculating text-acc1 and text-f1 scores.

**Efficiency of our Baselines.**   Our text-based evaluations were $\sim 7\times$ faster compared to the image dataset evaluations.

### B.1   Prompting Templates

We use the same prompting strategy introduced by Radford et al. [2021] to generate the model zero-shot weights (see Fig. 10 for examples of templates from different datasets). Specifically, for

Table 3: Details on the different benchmarks used in the study, including the number of classes, tasks, and target domain.

| Dataset | Classes | Task | Domain |
|---|---|---|---|
| Imagenet [Deng et al., 2009] | 1000 | classification | natural image |
| Stanford Cars [Krause et al., 2013] | 196 | classification | natural image |
| Flowers102 [Nilsback and Zisserman, 2008] | 102 | classification | natural image |
| CIFAR100 [Krizhevsky et al., 2009] | 100 | classification | natural image |
| GTSRB [Stallkamp et al., 2011] | 43 | classification | natural image |
| VOC2007 [Everingham et al., 2007] | 20 | classification | natural image |
| Oxford Pets [Parkhi et al., 2012] | 37 | classification | natural image |
| STL10 [Coates et al., 2011] | 10 | classification | natural image |
| DTD [Cimpoi et al., 2014] | 46 | classification | textural image |
| RESISC45 [Cheng et al., 2017] | 45 | classification | satellite images |
| EuroSAT [Helber et al., 2019] | 10 | classification | satellite images |
| MNIST [LeCun et al., 2010] | 10 | classification | hand-writing |
| Retinopathy [Kaggle and EyePacs, 2015] | 5 | classification | retina scan |
| PCam [Veeling et al., 2018] | 2 | classification | histopathology |
| SUN397 [Xiao et al., 2010] | 397 | scene und. | natural image |
| Country211 [Radford et al., 2021] | 211 | geolocation | natural image |
| SVHN [Netzer et al., 2011] | 10 | OCR | natural image |
| Rendered SST2 [Radford et al., 2021] | 2 | OCR | text image |
| FER2013 [Dumitru Ian Goodfellow, 2013] | 7 | fac. exp. rec. | natural image |
| CLEVR-C [Johnson et al., 2017] | 8 | object counting | natural image |
| CLEVR-D [Johnson et al., 2017] | 8 | distance est. | natural image |
| DMLab [Zhai et al., 2020] | 6 | distance est. | synthetic |
| KITTI [Geiger et al., 2013] | 4 | distance est. | natural image |

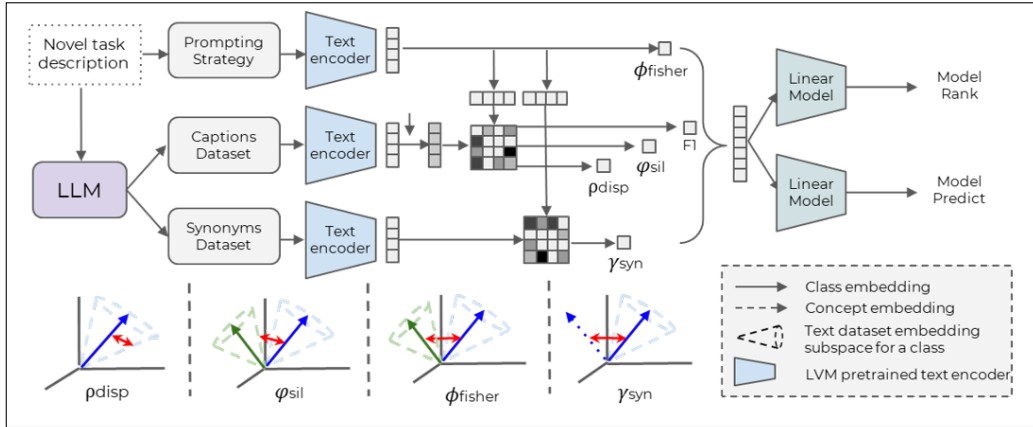

Figure 7: **Baselines Overview**. (top left) Using a text description of a new task, we use a large language model to generate the Image Caption and Class-Synonym datasets. We feed these text datasets into a VLMs text encoder, which generates the text-derived multi-modal embeddings. Using these embeddings, as well as the user-defined prompting strategies, we extract different scores. Finally, we fit a linear model on the extracted scores to predict model ranking and accuracy. (bottom) Schematic drawings of our different proposed scores.

every class $c$, we used the reported templates to produce the text prompts $\boldsymbol{Y}^c$ and encoded these prompts using the VLM text encoder, $f_y$, to produce the text embeddings for class $c$, $\hat{\boldsymbol{y}}^c$:

$$\hat{\boldsymbol{y}}^c = f_y(\boldsymbol{Y}^c).$$

We then normalized each separate prompt by its $L_2$ norm and averaged the resulting vector to produce $\bar{\boldsymbol{y}}^c$, or the unnormalized zero-shot weight of class $c$:

$$\bar{\boldsymbol{y}}^c = \frac{1}{N} \sum_{j=1}^{N} \frac{\boldsymbol{y}_j^c}{||\boldsymbol{y}_j^c||_2},$$

Table 4: **Translation of open clip to model/pre-training dataset names used in paper.** When renaming the datasets we tried to group models with similar optimization schemes to minimize the number of pre-training datasets without causing undo overlap.

| ID | Model | Name | Dataset | Name |
|----|-------|------|---------|------|
| 1 | RN50 | RN50 | openai | WIT |
| 2 | RN101 | RN101 | openai | WIT |
| 3 | RN50x4 | RN50x4 | openai | WIT |
| 4 | RN50-16 | RN50x16 | openai | WIT |
| 5 | RN50x64 | RN50x64 | openai | WIT |
| 6 | ViT-B-32 | ViT-B/32 | laion400m_e31 | L400m |
| 7 | ViT-B-32 | ViT-B/32 | laion400m_e32 | L400m |
| 8 | ViT-B-32-quickgelu | ViT-B/32 | laion400m_e32 | L400m |
| 9 | ViT-B-32 | ViT-B/32 | openai | WIT |
| 10 | ViT-B-32 | ViT-B/32 | laion2b_s34b_b79k | L2b-b |
| 11 | ViT-B-32 | ViT-B/32 | laion2b_e16 | L2b-c |
| 12 | ViT-B-16 | ViT-B/16 | laion400m_e32 | L400m |
| 13 | ViT-B-16 | ViT-B/16 | openai | WIT |
| 14 | ViT-B-16-240 | ViT-B/16-240 | laion400m_e32 | L400m |
| 15 | ViT-L-14 | ViT-L/14 | laion400m_e31 | L400m |
| 16 | ViT-L-14 | ViT-L/14 | laion400m_e32 | L400m |
| 17 | ViT-L-14 | ViT-L/14 | laion2b_s32b_b82k | L2b-b |
| 18 | ViT-L-14 | ViT-L/14 | openai | WIT |
| 19 | ViT-L-14-336 | ViT-L/14-336 | openai | WIT |
| 20 | ViT-G-14 | ViT-G/14 | laion2b_s12b_b42k | L2b-a |
| 21 | ViT-G-14 | ViT-G/14 | laion2b_s34b_b88k | L2b-a |
| 22 | ViT-H-14 | ViT-H/14 | laion2b_s32b_b79k | L2b-b |
| 23 | coca_ViT-B-32 | CoCa-ViT-B/32 | laion2b_s13b_b90k | L2b-c |
| 24 | coca_ViT-B-32 | CoCa-ViT-B/32 | mscoco_finetuned_laion2b_s13b_b90k | L2b-c + coco |
| 25 | coca_ViT-L-14 | CoCa-ViT-L/14 | laion2b_s13b_b90k | L2b-c |
| 26 | coca_ViT-L-14 | CoCa-ViT-L/14 | mscoco_finetuned_laion2b_s13b_b90k | L2b-c + coco |
| 27 | convnext_base | ConvNEXT-B | laion400m_s13b_b51k | L400m-c |
| 28 | convnext_base_w | ConvNEXT-BW | laion2b_s13b_b82k | L2b-d |
| 29 | convnext_base_w | ConvNEXT-BW | laion2b_s13b_b82k_augreg | L2b-e |
| 30 | convnext_base_w | ConvNEXT-BW | laion_aesthetic_s13b_b82k | L2b-f |
| 31 | convnext_base_w_320 | ConvNEXT-BW-320 | laion_aesthetic_s13b_b82k | L2b-f |
| 32 | convnext_base_w_320 | ConvNEXT-BW-320 | laion_aesthetic_s13b_b82k_augreg | L2b-g |
| 33 | convnext_large_d | ConvNEXT-LD | laion2b_s26b_b102k_augreg | L2b-h |
| 34 | convnext_large_d_320 | ConvNEXT-LD-320 | laion2b_s29b_b131k_ft | L2b-i |
| 35 | convnext_large_d_320 | ConvNEXT-LD-320 | laion2b_s29b_b131k_ft_soup | L2b-j |

where $\bar{y}^c$ is then normalized again to produce the final zero-shot classification weight of class $c$,

$$y^c = \frac{\bar{y}^c}{||\bar{y}^c||_2}. \tag{4}$$

## B.2   Text-Derived Scores

We define the six scores we derived for model selection and performance prediction. The *Text top-1 accuracy score* and *Text f1-score* is used to estimate the VLMs' performance on a vision task using text as a proxy, while the *Fisher criterion* and *Silhouette score* are used to understand the VLM's capability to separate samples from different classes in the target task (inter-class similarity. To estimate dataset granularity, we use *Class Dispersion score*. Finally, the *Synonym Consistency score* allows us to evaluate the degree of content shift between the VLMs' pre-training and target dataset (intra-class similarity).

**Text Classification scores.**   We use the generated captions dataset (see Sec. 3.1) as a proxy for images and evaluate the resulting model performance. Specifically, we use the VLM text encoder to generate text-derived multi-modal embeddings. We then corrupt these embeddings with Gaussian noise to approximate image-instance variation (see Sec. B.4)and calculate their cosine similarity with the class prompt embeddings - derived using the same prompt ensembling strategies proposed by Radford et al. [2021] (see Fig. 7). We then calculate the text top-1 accuracy (**text-acc1**) and text f1-score (**text-f1**).

**Fisher criterion,** $\phi_{\text{fisher}}$**.** The Fisher criterion [Fisher, 1936] has been widely used as a dataset granularity measure and has recently been shown to be effective for classification by Cui et al. [2019]. The Fisher score measures the degree of similarity of the classes or the extent of their separation. In VLMs, The quality of the class separation can be evaluated using text by assessing how close the different (text-derived) class prompt embeddings are. We introduce the concept of *Fisher criterion*, a score that quantifies how close the class prompt embeddings are to each other (see Fig. 7):

$$\phi_{\text{fisher}} = \frac{1}{C} \sum_{j=1}^{C} \max_{c, c \neq j} \left[ \theta(\boldsymbol{y}^j, \boldsymbol{y}^c) \right], \tag{5}$$

where $\boldsymbol{y}^c$ is the class prompt embedding derived using the prompt ensembling strategies proposed in Radford et al. [2021] for class $c$ (see Sec. B.1), $\theta(\cdot, \cdot)$ is a function that calculates the cosine similarity between two vectors, and $C$ is the number of classes.

**Silhouette score,** $\varphi_{\text{sil}}$**.** The silhouette score [Rousseeuw, 1987] is a well-established score that has been used to quantify the tightness of the same-class samples to the separation of different-class samples [Scheidegger et al., 2021, Cui et al., 2019]. Inspired by this score, we introduce the text-based *Silhouette score*, $\varphi_{\text{sil}}$, which measures the separation of different-class samples in the caption dataset $\boldsymbol{D}^{\text{cap}}$. To do so, we evaluate the average cosine similarity of captions to the nearest other class by:

$$\varphi_{\text{sil}} = \frac{1}{C} \sum_{j=1}^{C} \max_{c, c \neq j} \left[ \frac{1}{N} \sum_{k=1}^{N} \theta(\boldsymbol{D}^{\text{cap}}[j]_k, \boldsymbol{y}^c) \right], \tag{6}$$

where $\boldsymbol{y}^c$ is the class prompt embedding derived using the prompt ensembling strategies proposed in Radford et al. [2021] for class $c$ (see Sec. B.1), $\theta(\cdot, \cdot)$ is a function that calculates the cosine similarity between two vectors, and $C$ is the number of classes. $\boldsymbol{D}^{\text{cap}}[j]_k$ representing sample $k$ of class $j$ in the caption dataset $\boldsymbol{D}^{\text{cap}}$, and there is a total of $N$ such samples in for each class.

**Class Dispersion score,** $\rho_{\text{disp}}$**.** The *Class Dispersion score* is used as the normalization constant to generate the Fisher and Silhouette scores, and it quantifies the degree of same-class tightness or data cone radius (see Fig. 7).

$$\rho_{\text{disp}} = \frac{1}{CN} \sum_{c=1}^{C} \sum_{k=1}^{N} \theta(\boldsymbol{D}^{\text{cap}}[c]_k, \boldsymbol{y}^c), \tag{7}$$

where $\boldsymbol{y}^c$ is the class prompt embedding derived using the prompt ensembling strategies proposed in Radford et al. [2021] for class $c$ (see Sec. B.1), $\theta(\cdot, \cdot)$ is a function that calculates the cosine similarity between two vectors, and $C$ is the number of classes. $\boldsymbol{D}^{\text{cap}}[c]_k$ representing sample $k$ of class $c$ in the caption dataset $\boldsymbol{D}^{\text{cap}}$, and there is a total of $N$ such samples in for each class.

**Synonym Consistency score,** $\gamma_{\text{syn}}$**.** Synonym consistency has been shown in large language models to correlate with the degree of familiarity of the model with a particular concept [van der Lee et al., 2023]. Using the Synonym dataset, we compare the cosine similarity between the text embedding of each class and its corresponding synonyms. A high cosine similarity between the class and its corresponding synonyms/supercategories indicates that the model is aware of the semantic meaning of the class and is defined as:

$$\gamma_{\text{syn}} = \frac{1}{CN} \sum_{c=1}^{C} \sum_{k=1}^{N} \theta(\boldsymbol{D}^{\text{syn}}[c]_k, \boldsymbol{y}^c), \tag{8}$$

where $\boldsymbol{y}^c$ is the class prompt embedding derived using the prompt ensembling strategies proposed in Radford et al. [2021] for class $c$ (see Sec. B.1), $\theta(\cdot, \cdot)$ is a function that calculates the cosine similarity between two vectors, and $C$ is the number of classes. $\boldsymbol{D}^{\text{syn}}[c]_k$ representing sample $k$ of class $c$ in the synonym dataset $\boldsymbol{D}^{\text{syn}}$, and there is a total of $N$ such samples in for each class.

### B.3 Text Dataset Generation

To generate the Captions dataset, we used a large language model to generate realistic (but confusing) image captions. It was necessary to request confusing image captions to get sufficient variation in

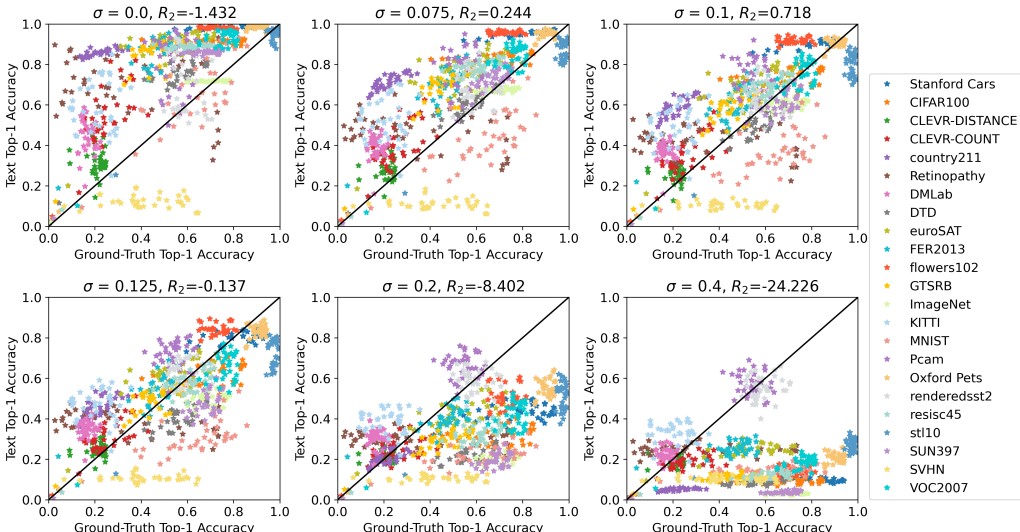

Figure 8: **Ablating Noise Injection Effect on Text Top-1 Accuracy**. Without noise (sigma=0), the text top-1 accuracy saturates on many datasets and models, with extremely high top-1 accuracy, making the correlation between the ground-truth top-1 accuracy and text top-1 accuracy quite poor. By corrupting the text embeddings with noise, we notice an improvement in correlation up to sigma=0.1, after which the correlation is steadily corrupted.

the image captions. We used OpenAI's 'gpt-3.5-turbo-0301' model with a temperature of 1. For the synonym dataset, we reduced the temperature to 0.1 and only requested the synonyms themselves. We then used the prompting templates with the synonym in place of the original class name to generate $D^{\text{syn}}$.

## B.4 Text Classification and Noise Corruption

In this work, we introduce the use of Gaussian noise to corrupt text-derived multi-modal embeddings to approximate image-instance variation. The corrupted embeddings are then used to calculate the text top-1 accuracy and text-f1 score, which serves as a proxy for evaluating the performance of a vision model. The scores are derived from the Captions dataset, a collection of complex but probable image captions generated using a large language model for images containing the user-provided classes in the user-provided domain and for the user-provided task. For more, see Sec. 3.1.

To evaluate the effectiveness of the text top-1 accuracy, we systematically increase the level of noise corruption and plot the text top-1 accuracy against the ground-truth top-1 accuracy (See Fig. 8). We quantify this correlation via the $R^2$ score or the degree of explained variance. As we do not fit a linear model to the predicted vs. ground-truth predictions, $R^2$ ranges from 1 (perfect linear fit) to $-\infty$, where non of the variance is explained. The results show that, without noise corruption, the text top-1 accuracy is too high and frequently saturates without any corruption. However, as the noise level increases to 0.1, the text top-1 accuracy progressively improves until a better linear correlation can be seen. This indicates that increasing noise corruption can better approximate image-instance variation and improve the correlation of the text top-1 accuracy. Beyond 0.1, however, the correlation between the text top-1 accuracy and top-1 accuracy progressively worsens. This shows that while noise corruption helps improve the correlation of the text top-1 accuracy, there is a limit beyond which further noise corruption degrades its effectiveness.

Table 5: **LOVM Benchmark (top-1 accuracy)**. We evaluate our method's performance over 23 datasets and 35 pre-trained models. (top) Model Ranking results. (bottom) Performance Prediction results. INB - ImageNet Benchmark score, C - Text Classification scores G - Granularity Scores.

| | | Stanford Cars | CIFAR100 | CLEVR-DIST. | CLEVR-COUNT | Country211 | Retinopathy | DMLab | DTD | EuroSAT | FER2013 | Flowers102 | GTSRB | ImageNet | KITTI | MNIST | PCam | Oxford Pets | Rendered SST2 | RESISC45 | STL10 | SUN397 | SVHN | VOC2007 | Mean |
|---|---|---|---|---|---|---|---|---|---|---|---|---|---|---|---|---|---|---|---|---|---|---|---|---|---|
| $R_5$ | INB | 0.80 | 0.80 | 0.00 | 0.60 | 0.40 | 0.00 | 0.00 | 1.00 | 0.60 | 0.20 | 0.40 | 0.20 | 1.00 | 0.00 | 0.20 | 0.00 | 0.80 | 0.00 | 1.00 | 0.20 | 0.60 | 0.60 | 0.60 | 0.452 |
| | C | 0.00 | 0.20 | 0.20 | 0.60 | 0.40 | 0.00 | 0.00 | 0.60 | 0.40 | 0.40 | 0.20 | 0.40 | 0.00 | 0.20 | 0.20 | 0.00 | 0.20 | 0.00 | 0.40 | 0.20 | 0.40 | 0.00 | 0.00 | 0.226 |
| | G | 0.40 | 0.40 | 0.20 | 0.20 | 0.40 | 0.00 | 0.00 | 0.40 | 0.20 | 0.20 | 0.80 | 0.20 | 0.40 | 0.20 | 0.00 | 0.20 | 0.40 | 0.00 | 0.40 | 0.20 | 0.40 | 0.00 | 0.20 | 0.252 |
| | G+C | 0.40 | 0.40 | 0.20 | 0.20 | 0.40 | 0.00 | 0.00 | 0.40 | 0.20 | 0.20 | 0.80 | 0.20 | 0.40 | 0.20 | 0.00 | 0.20 | 0.40 | 0.00 | 0.40 | 0.20 | 0.40 | 0.00 | 0.00 | 0.252 |
| | INB+C | 0.80 | 0.80 | 0.00 | 0.60 | 0.60 | 0.00 | 0.00 | 1.00 | 0.60 | 0.20 | 0.60 | 0.60 | 0.80 | 0.00 | 0.20 | 0.00 | 0.80 | 0.00 | 1.00 | 0.20 | 0.60 | 0.40 | 0.60 | 0.452 |
| | INB+G | 0.80 | 0.80 | 0.00 | 0.60 | 0.40 | 0.00 | 0.00 | 1.00 | 0.60 | 0.20 | 0.40 | 0.60 | 1.00 | 0.00 | 0.40 | 0.00 | 0.80 | 0.20 | 1.00 | 0.40 | 0.60 | 0.40 | 0.40 | **0.461** |
| | INB+C+G | 0.80 | 0.80 | 0.00 | 0.60 | 0.40 | 0.00 | 0.00 | 1.00 | 0.60 | 0.20 | 0.40 | 0.60 | 1.00 | 0.00 | 0.40 | 0.00 | 0.80 | 0.20 | 1.00 | 0.40 | 0.60 | 0.40 | 0.40 | **0.461** |
| $\tau$ | INB | 0.33 | 0.67 | 0.00 | 0.33 | 0.00 | 0.00 | 0.00 | -0.20 | 1.00 | 0.00 | 0.00 | 1.00 | 1.00 | 0.00 | 0.00 | 0.00 | 0.00 | 0.00 | -0.40 | 0.00 | -1.00 | 0.33 | 1.00 | 0.177 |
| | C | 0.00 | 0.00 | 0.00 | 1.00 | 0.00 | 0.00 | 0.00 | 0.33 | 0.00 | 0.00 | 0.00 | 0.00 | 0.00 | 0.00 | 0.00 | 0.00 | 0.00 | 0.00 | 0.00 | 0.00 | 0.00 | 0.00 | 0.00 | 0.058 |
| | G | 0.00 | 0.00 | 0.00 | 0.00 | 0.00 | 0.00 | 0.00 | 0.00 | 0.00 | 0.00 | 0.00 | -0.33 | 0.00 | 0.00 | 0.00 | 0.00 | 0.00 | 0.00 | 0.00 | 0.00 | 0.00 | 0.00 | 0.00 | -0.014 |
| | C+G | 0.00 | 0.00 | 0.00 | 0.00 | 0.00 | 0.00 | 0.00 | 0.00 | 0.00 | 0.00 | 0.00 | -0.33 | 0.00 | 0.00 | 0.00 | 0.00 | 0.00 | 0.00 | 0.00 | 0.00 | 0.00 | 0.00 | 0.00 | -0.014 |
| | INB+C | 0.33 | 0.67 | 0.00 | 0.33 | -0.33 | 0.00 | 0.00 | 0.00 | 1.00 | 0.00 | 1.00 | 1.00 | 1.00 | 0.00 | 0.00 | 0.00 | 0.33 | 0.00 | -0.20 | 0.00 | -1.00 | 0.00 | 1.00 | **0.223** |
| | INB+G | 0.33 | 0.33 | 0.00 | 0.33 | 0.00 | 0.00 | 0.00 | 0.00 | 1.00 | 0.00 | 0.00 | 1.00 | 0.80 | 0.00 | 0.00 | 0.00 | 0.00 | 0.00 | -0.60 | 0.00 | -1.00 | 0.00 | 0.00 | 0.096 |
| | INB+C+G | 0.33 | 0.33 | 0.00 | 0.33 | 0.00 | 0.00 | 0.00 | 0.00 | 1.00 | 0.00 | 0.00 | 1.00 | 0.80 | 0.00 | 0.00 | 0.00 | 0.00 | 0.00 | -0.60 | 0.00 | -1.00 | 0.00 | 0.00 | 0.096 |
| $L_1$ | INB | 0.16 | 0.07 | 0.56 | 0.42 | 0.49 | 0.30 | 0.62 | 0.09 | 0.08 | 0.25 | 0.01 | 0.23 | 0.00 | 0.63 | 0.04 | 0.26 | 0.17 | 0.18 | 0.07 | 0.21 | 0.03 | 0.17 | 0.03 | 0.220 |
| | C | 0.08 | 0.13 | 0.19 | 0.15 | 0.25 | 0.25 | 0.32 | 0.17 | 0.05 | 0.13 | 0.15 | 0.03 | 0.19 | 0.37 | 0.28 | 0.34 | 0.10 | 0.14 | 0.05 | 0.23 | 0.21 | 0.16 | 0.07 | 0.176 |
| | G | 0.03 | 0.03 | 0.19 | 0.22 | 0.44 | 0.16 | 0.23 | 0.01 | 0.03 | 0.07 | 0.45 | 0.00 | 0.22 | 0.29 | 0.29 | 0.04 | 0.09 | 0.11 | 0.02 | 0.21 | 0.03 | 0.11 | 0.02 | 0.144 |
| | C+G | 0.03 | 0.03 | 0.19 | 0.22 | 0.44 | 0.16 | 0.23 | 0.01 | 0.03 | 0.07 | 0.45 | 0.00 | 0.22 | 0.29 | 0.29 | 0.04 | 0.09 | 0.11 | 0.02 | 0.21 | 0.03 | 0.11 | 0.02 | 0.144 |
| | INB+C | 0.08 | 0.13 | 0.19 | 0.15 | 0.25 | 0.25 | 0.32 | 0.17 | 0.05 | 0.13 | 0.15 | 0.03 | 0.19 | 0.37 | 0.28 | 0.34 | 0.10 | 0.14 | 0.05 | 0.23 | 0.21 | 0.16 | 0.07 | 0.176 |
| | INB+G | 0.03 | 0.02 | 0.20 | 0.18 | 0.44 | 0.02 | 0.28 | 0.01 | 0.02 | 0.10 | 0.43 | 0.02 | 0.21 | 0.33 | 0.25 | 0.03 | 0.06 | 0.12 | 0.00 | 0.23 | 0.03 | 0.22 | 0.00 | **0.140** |
| | INB+C+G | 0.03 | 0.02 | 0.20 | 0.18 | 0.44 | 0.02 | 0.28 | 0.01 | 0.02 | 0.10 | 0.43 | 0.02 | 0.21 | 0.33 | 0.25 | 0.03 | 0.06 | 0.12 | 0.00 | 0.23 | 0.03 | 0.22 | 0.00 | **0.140** |

Table 6: **LOVM Benchmark (mean per-class recall)**. We evaluate our method's performance over 23 datasets and 35 pre-trained models. (top) Model Ranking results. (bottom) Performance Prediction results. INB - ImageNet Benchmark score, C - Text Classification scores G - Granularity Scores.

| | | Stanford Cars | CIFAR100 | CLEVR-DIST. | CLEVR-COUNT | Country211 | Retinopathy | DMLab | DTD | EuroSAT | FER2013 | Flowers102 | GTSRB | ImageNet | KITTI | MNIST | PCam | Oxford Pets | Rendered SST2 | RESISC45 | STL10 | SUN397 | SVHN | VOC2007 | Mean |
|---|---|---|---|---|---|---|---|---|---|---|---|---|---|---|---|---|---|---|---|---|---|---|---|---|---|
| $R_5$ | INB | 0.80 | 0.80 | 0.40 | 0.60 | 0.40 | 0.40 | 0.00 | 1.00 | 0.60 | 0.40 | 0.60 | 0.60 | 1.00 | 0.20 | 0.20 | 0.00 | 0.60 | 0.00 | 1.00 | 0.20 | 0.80 | 0.60 | 0.40 | 0.504 |
| | C | 0.00 | 0.20 | 0.20 | 0.60 | 0.40 | 0.40 | 0.20 | 0.20 | 0.60 | 0.40 | 0.40 | 0.00 | 0.40 | 0.00 | 0.40 | 0.20 | 0.20 | 0.20 | 0.20 | 0.40 | 0.20 | 0.40 | 0.00 | 0.252 |
| | G | 0.40 | 0.40 | 0.20 | 0.20 | 0.40 | 0.00 | 0.40 | 0.40 | 0.20 | 0.20 | 0.60 | 0.20 | 0.40 | 0.40 | 0.00 | 0.40 | 0.40 | 0.00 | 0.40 | 0.20 | 0.40 | 0.20 | 0.00 | 0.270 |
| | G+C | 0.40 | 0.40 | 0.20 | 0.20 | 0.40 | 0.00 | 0.40 | 0.40 | 0.20 | 0.20 | 0.60 | 0.20 | 0.40 | 0.40 | 0.00 | 0.40 | 0.40 | 0.00 | 0.40 | 0.20 | 0.40 | 0.20 | 0.00 | 0.270 |
| | INB+C | 0.80 | 0.80 | 0.40 | 0.60 | 0.60 | 0.40 | 0.00 | 1.00 | 0.60 | 0.40 | 0.80 | 0.60 | 0.80 | 0.20 | 0.20 | 0.00 | 0.80 | 0.00 | 1.00 | 0.20 | 0.80 | 0.40 | 0.40 | 0.513 |
| | INB+G | 0.80 | 0.80 | 0.40 | 0.60 | 0.40 | 0.40 | 0.00 | 1.00 | 0.60 | 0.40 | 0.60 | 0.60 | 1.00 | 0.20 | 0.40 | 0.20 | 0.60 | 0.20 | 1.00 | 0.40 | 0.80 | 0.60 | 0.60 | **0.548** |
| | INB+C+G | 0.80 | 0.80 | 0.40 | 0.60 | 0.40 | 0.40 | 0.00 | 1.00 | 0.60 | 0.40 | 0.60 | 0.60 | 1.00 | 0.20 | 0.40 | 0.20 | 0.60 | 0.20 | 1.00 | 0.40 | 0.80 | 0.60 | 0.60 | **0.548** |
| $\tau$ | INB | 0.67 | 0.67 | 0.00 | 0.33 | 0.00 | 0.00 | 0.00 | 0.00 | 1.00 | 0.00 | 1.00 | 1.00 | 1.00 | 0.00 | 0.00 | 0.00 | -0.33 | 0.00 | -0.40 | 0.00 | -0.33 | -0.33 | 0.00 | 0.186 |
| | C | 0.00 | 0.00 | 0.00 | 1.00 | 0.00 | 0.00 | 0.00 | 0.33 | 0.00 | 0.00 | 0.00 | 0.00 | 0.00 | 0.00 | 0.00 | 0.00 | 0.00 | 0.00 | 0.00 | 0.00 | 0.00 | 0.00 | 0.00 | 0.058 |
| | G | 0.00 | 0.00 | 0.00 | 0.00 | 0.00 | 0.00 | 0.00 | 0.00 | 0.00 | 0.00 | 0.00 | -0.33 | 0.00 | 0.00 | 0.00 | 0.00 | 0.00 | 0.00 | 0.00 | 0.00 | 0.00 | 0.00 | 0.00 | -0.014 |
| | C+G | 0.00 | 0.00 | 0.00 | 0.00 | 0.00 | 0.00 | 0.00 | 0.00 | 0.00 | 0.00 | 0.00 | -0.33 | 0.00 | 0.00 | 0.00 | 0.00 | 0.00 | 0.00 | 0.00 | 0.00 | 0.00 | 0.00 | 0.00 | -0.014 |
| | INB+C | 0.67 | 0.67 | 0.00 | 0.33 | -0.33 | 0.00 | 0.00 | -0.20 | 1.00 | 0.00 | 1.00 | 1.00 | 0.67 | 0.00 | 0.00 | 0.00 | 0.00 | 0.00 | -0.20 | 0.00 | 0.00 | 0.00 | 0.00 | **0.200** |
| | INB+G | 0.67 | 0.33 | 0.00 | 0.33 | 0.00 | 0.00 | 0.00 | 0.00 | 1.00 | 0.00 | 1.00 | 1.00 | 0.80 | 0.00 | 0.00 | 0.00 | 0.33 | 0.00 | -0.60 | 0.00 | -0.33 | -0.33 | 0.33 | 0.197 |
| | INB+C+G | 0.67 | 0.33 | 0.00 | 0.33 | 0.00 | 0.00 | 0.00 | 0.00 | 1.00 | 0.00 | 1.00 | 1.00 | 0.80 | 0.00 | 0.00 | 0.00 | 0.33 | 0.00 | -0.60 | 0.00 | -0.33 | -0.33 | 0.33 | 0.197 |
| $L_1$ | INB | 0.16 | 0.07 | 0.58 | 0.42 | 0.49 | 0.52 | 0.61 | 0.09 | 0.08 | 0.26 | 0.01 | 0.26 | 0.00 | 0.50 | 0.03 | 0.26 | 0.17 | 0.18 | 0.06 | 0.21 | 0.03 | 0.17 | 0.08 | 0.228 |
| | C | 0.07 | 0.13 | 0.21 | 0.15 | 0.25 | 0.29 | 0.31 | 0.18 | 0.04 | 0.16 | 0.17 | 0.06 | 0.20 | 0.30 | 0.29 | 0.35 | 0.09 | 0.15 | 0.05 | 0.23 | 0.21 | 0.18 | 0.12 | 0.182 |
| | G | 0.14 | 0.03 | 0.25 | 0.16 | 0.44 | 0.23 | 0.19 | 0.01 | 0.01 | 0.12 | 0.46 | 0.03 | 0.30 | 0.15 | 0.28 | 0.03 | 0.08 | 0.09 | 0.04 | 0.16 | 0.02 | 0.01 | 0.02 | **0.141** |
| | C+G | 0.14 | 0.03 | 0.25 | 0.16 | 0.44 | 0.23 | 0.19 | 0.01 | 0.01 | 0.12 | 0.46 | 0.03 | 0.30 | 0.15 | 0.28 | 0.03 | 0.08 | 0.09 | 0.04 | 0.16 | 0.02 | 0.01 | 0.02 | **0.141** |
| | INB+C | 0.07 | 0.13 | 0.21 | 0.15 | 0.25 | 0.29 | 0.31 | 0.18 | 0.04 | 0.16 | 0.17 | 0.06 | 0.20 | 0.30 | 0.29 | 0.35 | 0.09 | 0.15 | 0.05 | 0.23 | 0.21 | 0.18 | 0.12 | 0.182 |
| | INB+G | 0.14 | 0.03 | 0.25 | 0.16 | 0.44 | 0.23 | 0.19 | 0.01 | 0.01 | 0.12 | 0.46 | 0.03 | 0.30 | 0.15 | 0.28 | 0.03 | 0.08 | 0.09 | 0.04 | 0.16 | 0.02 | 0.01 | 0.02 | **0.141** |
| | INB+C+G | 0.14 | 0.03 | 0.25 | 0.16 | 0.44 | 0.23 | 0.19 | 0.01 | 0.01 | 0.12 | 0.46 | 0.03 | 0.30 | 0.15 | 0.28 | 0.03 | 0.08 | 0.09 | 0.04 | 0.16 | 0.02 | 0.01 | 0.02 | **0.141** |

# C  Additional Results

## C.1  LOVM Per-Dataset Breakdown

Here, we show the per-dataset breakdown of our main results. In Tab. 5, we show our model ranking and performance prediction results for top-1 accuracy. In Tab. 6, we show our model ranking and performance prediction results for mean per-class recall.

Table 7: **Effect of Using Different LLM**. This table shows the results of re-running our baselines with a different LLM (gpt-3.5-turbo-16k instead of gpt-3.5-turbo-0301). We evaluate mean per-class recall and top-1 accuracy for various combinations of datasets and models. INB - ImageNet Baseline, C - Text Classification scores, G - Granularity scores.

| Used Scores | Mean Per-Class Recall | | | Top-1 Accuracy | | |
|---|---|---|---|---|---|---|
| | $R_5(\uparrow)$ | $\tau(\uparrow)$ | $L_1(\downarrow)$ | $R_5(\uparrow)$ | $\tau(\uparrow)$ | $L_1(\downarrow)$ |
| INB | 0.504 | 0.186 | 0.228 | 0.452 | 0.177 | 0.220 |
| C | 0.365 | 0.072 | 0.144 | 0.357 | 0.043 | 0.145 |
| G | 0.252 | 0.014 | **0.133** | 0.243 | 0.029 | 0.135 |
| G+C | 0.365 | 0.072 | **0.133** | 0.357 | 0.043 | **0.129** |
| INB+C | 0.504 | **0.223** | 0.144 | 0.461 | 0.200 | 0.145 |
| INB+G | **0.522** | 0.191 | **0.133** | 0.461 | **0.212** | 0.135 |
| INB+G+C | **0.522** | 0.191 | **0.133** | **0.470** | 0.078 | **0.129** |

## C.2  Ablation Experiments

To understand the utility of each of our extracted scores, we exhaustively ablated their effect on top-1 accuracy model ranking and performance prediction (Tab. 8 and Tab. 9), and mean per-class recall model ranking and performance prediction (Tab. 10 and Tab. 11). Specifically, we ablated each score's impact on the resulting model's performance. As can be seen, using more than $\sim 3$ features at a time seldom improves performance. Future work can investigate the use of more sophisticated models that may be able to utilize more scores in predicting model ranking and performance. Specifically, for ranking models, text classification and scores quantifying intra-class similarity (Class Dispersion score & Synonym Consistency score) were the most dominant, while for performance prediction, granularity scores quantifying both inter- and inta- class similarity was the most important. This

We ablate the model ranking performance to understand each extracted score's effect on ranking models. The text classification scores and scores quantifying intra-class similarity were the most consequential in predicting model ranking. Specifically, in ranking models, the text-f1 score, Class Dispersion score ($\rho_{disp}$), and Synonym Consistency score ($\gamma_{syn}$) where the most dominant (Tab. 8 rows 8 & 11, Tab. 10 row 38). Overall, it seems like the text classification excelled at fine-grained ranking (as quantified by $\tau$), while the inter-class granularity scores improved the coarse ranking prediction (as quantified by $R_5$. Meanwhile, granularity scores quantifying inter- and intra- class similarity were the most dominant for performance prediction. Specifically, Class Dispersion score ($\rho_{disp}$), Synonym Consistency score ($\gamma_{syn}$), and Silhouette score ($\varphi_{sil}$) were the most influential (Tab. 9 rows 26 & 41, Tab. 11 row 60). INB does not aid performance prediction, indicating that getting a course estimation of dataset difficulty dominates performance prediction.

## C.3  Large Language Model Ablation

We re-ran our experiments with a different large language model (gpt-3.5-turbo-16k) to assess the impact of the choice Large Language Model (LLM) on our results. Tab. 7 presents the outcomes of these experiments, including mean per-class recall and top-1 accuracy across various datasets and model combinations. Notably, while some variations in performance metrics were observed, it is important to emphasize that these variations do not substantially alter the overall conclusions of our study. Our findings consistently demonstrate the effectiveness of LOVM, irrespective of the specific LLM used for text dataset generation.

## C.4  Raw Model Ranking Details

To illustrate the model ranking of the naive (ImageNet Benchmark) baseline to some text-based approaches, we visualize the raw ranking prediction of each method. We sort the datasets from natural image classification (Fig. 9 left) to non-natural image / non-classification benchmarks (Fig. 9 right). Here, the evident failure of the ImageNet Benchmark baseline to capture dataset-specific changes in ranking is apparent. As the benchmark approach ranks models by their ImageNet performance, the model ranking is constant for all datasets. Meanwhile, integrating the text features produces a ranking distribution with a discernible positive correlation between the ground truth and the predicted model

Table 8: **LOVM Model Selection Ablation.** Here, we ablate all the different scores used in our baselines for model ranking by top-1 accuracy. We separated the text classification (C) base scores, and the granularity-based scores that quantify inter- and intra-class similarity. $a_{IN}$ - Imagenet Accuracy, $\phi_{fisher}$ - Fisher criterion, text-f1 - caption dataset f1-score, text-acc1 - text top-1 accuracy, $\gamma_{syn}$ - Synonym Consistency score, $\varphi_{sil}$ - Silhouette score, $\rho_{disp}$ - Class Dispersion score.

| | | | Scores | | | | | Metrics | |
| Row ID | $a_{IN}$ | text-f1 | text-acc1 | $\gamma_{syn}$ | $\rho_{disp}$ | $\varphi_{sil}$ | $\phi_{fisher}$ | $\tau$ (↑) | $R_5$ (↑) |
|---|---|---|---|---|---|---|---|---|---|
| 1 | ✓ | × | × | × | × | × | × | 0.177 | 0.452 |
| 2 | × | ✓ | × | × | × | × | × | 0.058 | 0.226 |
| 3 | × | × | ✓ | × | × | × | × | 0.029 | 0.191 |
| 4 | × | × | × | ✓ | × | × | × | 0.000 | 0.183 |
| 5 | × | × | × | × | ✓ | × | × | -0.014 | 0.243 |
| 6 | × | × | × | × | × | ✓ | × | -0.014 | 0.252 |
| 7 | × | × | × | × | × | × | ✓ | 0.000 | 0.165 |
| 8 | ✓ | ✓ | × | × | × | × | × | 0.223 | 0.452 |
| 9 | ✓ | × | ✓ | × | × | × | × | 0.200 | 0.452 |
| 10 | ✓ | × | × | ✓ | × | × | × | 0.188 | 0.426 |
| 11 | ✓ | × | × | × | ✓ | × | × | 0.096 | 0.461 |
| 12 | ✓ | × | × | × | × | ✓ | × | 0.078 | 0.417 |
| 13 | ✓ | × | × | × | × | × | ✓ | 0.110 | 0.426 |
| 14 | × | ✓ | ✓ | × | × | × | × | 0.029 | 0.226 |
| 15 | × | ✓ | × | ✓ | × | × | × | 0.043 | 0.191 |
| 16 | × | ✓ | × | × | ✓ | × | × | 0.014 | 0.209 |
| 17 | × | ✓ | × | × | × | ✓ | × | 0.014 | 0.217 |
| 18 | × | ✓ | × | × | × | × | ✓ | 0.000 | 0.183 |
| 19 | × | × | ✓ | ✓ | × | × | × | 0.043 | 0.191 |
| 20 | × | × | ✓ | × | ✓ | × | × | 0.014 | 0.209 |
| 21 | × | × | ✓ | × | × | ✓ | × | 0.014 | 0.191 |
| 22 | × | × | ✓ | × | × | × | ✓ | 0.000 | 0.174 |
| 23 | × | × | × | ✓ | ✓ | × | × | 0.000 | 0.200 |
| 24 | × | × | × | ✓ | × | ✓ | × | 0.014 | 0.235 |
| 25 | × | × | × | ✓ | × | × | ✓ | 0.000 | 0.174 |
| 26 | × | × | × | × | ✓ | ✓ | × | 0.014 | 0.209 |
| 27 | × | × | × | × | ✓ | × | ✓ | 0.000 | 0.165 |
| 28 | × | × | × | × | × | ✓ | ✓ | 0.014 | 0.200 |
| 29 | ✓ | ✓ | ✓ | × | × | × | × | 0.107 | 0.443 |
| 30 | ✓ | ✓ | × | ✓ | × | × | × | 0.174 | 0.443 |
| 31 | ✓ | ✓ | × | × | ✓ | × | × | 0.188 | 0.435 |
| 32 | ✓ | ✓ | × | × | × | ✓ | × | 0.145 | 0.443 |
| 33 | ✓ | ✓ | × | × | × | × | ✓ | 0.096 | 0.417 |
| 34 | ✓ | × | ✓ | ✓ | × | × | × | 0.159 | 0.435 |
| 35 | ✓ | × | ✓ | × | ✓ | × | × | 0.188 | 0.443 |
| 36 | ✓ | × | ✓ | × | × | ✓ | × | 0.116 | 0.443 |
| 37 | ✓ | × | ✓ | × | × | × | ✓ | 0.110 | 0.426 |
| 38 | ✓ | × | × | ✓ | ✓ | × | × | 0.110 | 0.426 |
| 39 | ✓ | × | × | ✓ | × | ✓ | × | 0.188 | 0.417 |
| 40 | ✓ | × | × | ✓ | × | × | ✓ | 0.110 | 0.417 |
| 41 | ✓ | × | × | × | ✓ | ✓ | × | 0.067 | 0.435 |
| 42 | ✓ | × | × | × | ✓ | × | ✓ | 0.110 | 0.426 |
| 43 | ✓ | × | × | × | × | ✓ | ✓ | 0.110 | 0.426 |
| 44 | × | ✓ | ✓ | ✓ | × | × | × | 0.000 | 0.209 |
| 45 | × | ✓ | ✓ | × | ✓ | × | × | 0.014 | 0.200 |
| 46 | × | ✓ | ✓ | × | × | ✓ | × | 0.014 | 0.200 |
| 47 | × | ✓ | ✓ | × | × | × | ✓ | 0.014 | 0.209 |
| 48 | × | ✓ | × | ✓ | ✓ | × | × | 0.000 | 0.217 |
| 49 | × | ✓ | × | ✓ | × | ✓ | × | 0.014 | 0.217 |
| 50 | × | ✓ | × | ✓ | × | × | ✓ | 0.000 | 0.191 |
| 51 | × | ✓ | × | × | ✓ | ✓ | × | 0.014 | 0.209 |
| 52 | × | ✓ | × | × | ✓ | × | ✓ | 0.014 | 0.217 |
| 53 | × | ✓ | × | × | × | ✓ | ✓ | 0.014 | 0.209 |
| 54 | × | × | ✓ | ✓ | ✓ | × | × | 0.014 | 0.217 |
| 55 | × | × | ✓ | ✓ | × | ✓ | × | 0.014 | 0.209 |
| 56 | × | × | ✓ | ✓ | × | × | ✓ | 0.043 | 0.191 |
| 57 | × | × | ✓ | × | ✓ | ✓ | × | 0.014 | 0.191 |
| 58 | × | × | ✓ | × | ✓ | × | ✓ | 0.014 | 0.200 |
| 59 | × | × | ✓ | × | × | ✓ | ✓ | 0.014 | 0.200 |
| 60 | × | × | × | ✓ | ✓ | ✓ | × | 0.014 | 0.200 |
| 61 | × | × | × | ✓ | ✓ | × | ✓ | 0.000 | 0.165 |
| 62 | × | × | × | ✓ | × | ✓ | ✓ | 0.014 | 0.191 |
| 63 | × | × | × | × | ✓ | ✓ | ✓ | 0.014 | 0.191 |
| 64 | ✓ | ✓ | ✓ | ✓ | × | × | × | 0.151 | 0.452 |
| 65 | ✓ | ✓ | ✓ | × | ✓ | × | × | 0.107 | 0.443 |
| 66 | ✓ | ✓ | ✓ | × | × | ✓ | × | 0.064 | 0.435 |
| 67 | ✓ | ✓ | ✓ | × | × | × | ✓ | 0.133 | 0.443 |
| 68 | ✓ | ✓ | × | ✓ | ✓ | × | × | 0.145 | 0.426 |
| 69 | ✓ | ✓ | × | ✓ | × | ✓ | × | 0.157 | 0.443 |
| 70 | ✓ | ✓ | × | ✓ | × | × | ✓ | 0.075 | 0.417 |
| 71 | ✓ | ✓ | × | × | ✓ | ✓ | × | 0.113 | 0.452 |
| 72 | ✓ | ✓ | × | × | ✓ | × | ✓ | 0.096 | 0.417 |
| 73 | ✓ | ✓ | × | × | × | ✓ | ✓ | 0.096 | 0.417 |
| 74 | ✓ | × | ✓ | ✓ | ✓ | × | × | 0.159 | 0.435 |
| 75 | ✓ | × | ✓ | ✓ | × | ✓ | × | 0.128 | 0.443 |
| 76 | ✓ | × | ✓ | ✓ | × | × | ✓ | 0.075 | 0.417 |
| 77 | ✓ | × | ✓ | × | ✓ | ✓ | × | 0.113 | 0.452 |
| 78 | ✓ | × | ✓ | × | ✓ | × | ✓ | 0.139 | 0.426 |
| 79 | ✓ | × | ✓ | × | × | ✓ | ✓ | 0.139 | 0.426 |
| 80 | ✓ | × | × | ✓ | ✓ | ✓ | × | 0.058 | 0.409 |
| 81 | ✓ | × | × | ✓ | ✓ | × | ✓ | 0.110 | 0.417 |
| 82 | ✓ | × | × | ✓ | × | ✓ | ✓ | 0.110 | 0.426 |
| 83 | ✓ | × | × | × | ✓ | ✓ | ✓ | 0.101 | 0.426 |
| 84 | × | ✓ | ✓ | ✓ | ✓ | × | × | -0.029 | 0.209 |
| 85 | × | ✓ | ✓ | ✓ | × | ✓ | × | -0.029 | 0.217 |
| 86 | × | ✓ | ✓ | ✓ | × | × | ✓ | -0.014 | 0.217 |
| 87 | × | ✓ | ✓ | × | ✓ | ✓ | × | 0.014 | 0.200 |
| 88 | × | ✓ | ✓ | × | ✓ | × | ✓ | 0.014 | 0.226 |
| 89 | × | ✓ | ✓ | × | × | ✓ | ✓ | 0.014 | 0.217 |
| 90 | × | ✓ | × | ✓ | ✓ | ✓ | × | 0.014 | 0.217 |
| 91 | × | ✓ | × | ✓ | ✓ | × | ✓ | 0.000 | 0.226 |
| 92 | × | ✓ | × | ✓ | × | ✓ | ✓ | 0.014 | 0.209 |
| 93 | × | ✓ | × | × | ✓ | ✓ | ✓ | 0.014 | 0.209 |
| 94 | × | × | ✓ | ✓ | ✓ | ✓ | × | 0.014 | 0.209 |
| 95 | × | × | ✓ | ✓ | ✓ | × | ✓ | 0.014 | 0.209 |
| 96 | × | × | ✓ | ✓ | × | ✓ | ✓ | 0.014 | 0.209 |
| 97 | × | × | ✓ | × | ✓ | ✓ | ✓ | 0.014 | 0.200 |
| 98 | × | × | × | ✓ | ✓ | ✓ | ✓ | 0.014 | 0.226 |
| 99 | ✓ | ✓ | ✓ | ✓ | ✓ | × | × | 0.072 | 0.426 |
| 100 | ✓ | ✓ | ✓ | ✓ | × | ✓ | × | 0.058 | 0.435 |
| 101 | ✓ | ✓ | ✓ | ✓ | × | × | ✓ | 0.133 | 0.443 |
| 102 | ✓ | ✓ | ✓ | × | ✓ | ✓ | × | 0.078 | 0.443 |
| 103 | ✓ | ✓ | ✓ | × | ✓ | × | ✓ | 0.090 | 0.452 |
| 104 | ✓ | ✓ | ✓ | × | × | ✓ | ✓ | 0.133 | 0.443 |
| 105 | ✓ | ✓ | × | ✓ | ✓ | ✓ | × | 0.119 | 0.443 |
| 106 | ✓ | ✓ | × | ✓ | ✓ | × | ✓ | 0.104 | 0.417 |
| 107 | ✓ | ✓ | × | ✓ | × | ✓ | ✓ | 0.104 | 0.417 |
| 108 | ✓ | ✓ | × | × | ✓ | ✓ | ✓ | 0.128 | 0.435 |
| 109 | ✓ | × | ✓ | ✓ | ✓ | ✓ | × | 0.128 | 0.443 |
| 110 | ✓ | × | ✓ | ✓ | ✓ | × | ✓ | 0.104 | 0.417 |
| 111 | ✓ | × | ✓ | ✓ | × | ✓ | ✓ | 0.104 | 0.417 |
| 112 | ✓ | × | ✓ | × | ✓ | ✓ | ✓ | 0.128 | 0.435 |
| 113 | ✓ | × | × | ✓ | ✓ | ✓ | ✓ | 0.101 | 0.400 |
| 114 | × | ✓ | ✓ | ✓ | ✓ | ✓ | × | -0.029 | 0.217 |
| 115 | × | ✓ | ✓ | ✓ | ✓ | × | ✓ | -0.043 | 0.209 |
| 116 | × | ✓ | ✓ | ✓ | × | ✓ | ✓ | -0.029 | 0.217 |
| 117 | × | ✓ | ✓ | × | ✓ | ✓ | ✓ | 0.014 | 0.217 |
| 118 | × | ✓ | × | ✓ | ✓ | ✓ | ✓ | 0.014 | 0.209 |
| 119 | × | × | ✓ | ✓ | ✓ | ✓ | ✓ | 0.014 | 0.209 |
| 120 | ✓ | ✓ | ✓ | ✓ | ✓ | ✓ | × | 0.055 | 0.443 |
| 121 | ✓ | ✓ | ✓ | ✓ | ✓ | × | ✓ | 0.061 | 0.435 |
| 122 | ✓ | ✓ | ✓ | ✓ | × | ✓ | ✓ | 0.090 | 0.435 |
| 123 | ✓ | ✓ | ✓ | × | ✓ | ✓ | ✓ | 0.090 | 0.443 |
| 124 | ✓ | ✓ | × | ✓ | ✓ | ✓ | ✓ | 0.128 | 0.435 |
| 125 | ✓ | × | ✓ | ✓ | ✓ | ✓ | ✓ | 0.128 | 0.435 |
| 126 | × | ✓ | ✓ | ✓ | ✓ | ✓ | ✓ | -0.029 | 0.217 |
| 127 | ✓ | ✓ | ✓ | ✓ | ✓ | ✓ | ✓ | 0.046 | 0.426 |

(Model Ranking)

Table 9: **LOVM Model Prediction Ablation.** Here, we ablate all the different scores used in our baselines for predicting model top-1 accuracy. We separated the text classification (C) base scores, and the granularity-based scores that quantify inter- and intra-class similarity. $a_{IN}$ - Imagenet Accuracy, $\phi_{fisher}$ - Fisher criterion, text-f1 - caption dataset f1-score, text-acc1 - text top-1 accuracy, $\gamma_{syn}$ - Synonym Consistency score, $\varphi_{sil}$ - Silhouette score, $\rho_{disp}$ - Class Dispersion score.

**Model Prediction**

| Row ID | $a_{IN}$ | text-f1 | text-acc1 | $\gamma_{syn}$ | $\rho_{disp}$ | $\varphi_{sil}$ | $\phi_{fisher}$ | $L_1(\downarrow)$ |
|---|---|---|---|---|---|---|---|---|
| 1 | ✓ | × | × | × | × | × | × | 0.220 |
| 2 | × | ✓ | × | × | × | × | × | 0.176 |
| 3 | × | × | ✓ | × | × | × | × | 0.177 |
| 4 | × | × | × | ✓ | × | × | × | 0.188 |
| 5 | × | × | × | × | ✓ | × | × | 0.200 |
| 6 | × | × | × | × | × | ✓ | × | 0.222 |
| 7 | × | × | × | × | × | × | ✓ | 0.170 |
| 8 | ✓ | ✓ | × | × | × | × | × | 0.189 |
| 9 | ✓ | × | ✓ | × | × | × | × | 0.184 |
| 10 | ✓ | × | × | ✓ | × | × | × | 0.215 |
| 11 | ✓ | × | × | × | ✓ | × | × | 0.205 |
| 12 | ✓ | × | × | × | × | ✓ | × | 0.215 |
| 13 | ✓ | × | × | × | × | × | ✓ | 0.168 |
| 14 | × | ✓ | ✓ | × | × | × | × | 0.179 |
| 15 | × | ✓ | × | ✓ | × | × | × | 0.190 |
| 16 | × | ✓ | × | × | ✓ | × | × | 0.186 |
| 17 | × | ✓ | × | × | × | ✓ | × | 0.180 |
| 18 | × | ✓ | × | × | × | × | ✓ | 0.160 |
| 19 | × | × | ✓ | ✓ | × | × | × | 0.183 |
| 20 | × | × | ✓ | × | ✓ | × | × | 0.177 |
| 21 | × | × | ✓ | × | × | ✓ | × | 0.179 |
| 22 | × | × | ✓ | × | × | × | ✓ | 0.159 |
| 23 | × | × | × | ✓ | ✓ | × | × | 0.199 |
| 24 | × | × | × | ✓ | × | ✓ | × | 0.197 |
| 25 | × | × | × | ✓ | × | × | ✓ | 0.154 |
| 26 | × | × | × | × | ✓ | ✓ | × | 0.144 |
| 27 | × | × | × | × | ✓ | × | ✓ | 0.163 |
| 28 | × | × | × | × | × | ✓ | ✓ | 0.165 |
| 29 | ✓ | ✓ | ✓ | × | × | × | × | 0.191 |
| 30 | ✓ | ✓ | × | ✓ | × | × | × | 0.197 |
| 31 | ✓ | ✓ | × | × | ✓ | × | × | 0.189 |
| 32 | ✓ | ✓ | × | × | × | ✓ | × | 0.187 |
| 33 | ✓ | ✓ | × | × | × | × | ✓ | 0.159 |
| 34 | ✓ | × | ✓ | ✓ | × | × | × | 0.196 |
| 35 | ✓ | × | ✓ | × | ✓ | × | × | 0.189 |
| 36 | ✓ | × | ✓ | × | × | ✓ | × | 0.185 |
| 37 | ✓ | × | ✓ | × | × | × | ✓ | 0.156 |
| 38 | ✓ | × | × | ✓ | ✓ | × | × | 0.212 |
| 39 | ✓ | × | × | ✓ | × | ✓ | × | 0.220 |
| 40 | ✓ | × | × | ✓ | × | × | ✓ | 0.156 |
| 41 | ✓ | × | × | × | ✓ | ✓ | × | 0.140 |
| 42 | ✓ | × | × | × | ✓ | × | ✓ | 0.160 |
| 43 | ✓ | × | × | × | × | ✓ | ✓ | 0.166 |
| 44 | × | ✓ | ✓ | ✓ | × | × | × | 0.184 |
| 45 | × | ✓ | ✓ | × | ✓ | × | × | 0.187 |
| 46 | × | ✓ | ✓ | × | × | ✓ | × | 0.186 |
| 47 | × | ✓ | ✓ | × | × | × | ✓ | 0.170 |
| 48 | × | ✓ | × | ✓ | ✓ | × | × | 0.190 |
| 49 | × | ✓ | × | ✓ | × | ✓ | × | 0.189 |
| 50 | × | ✓ | × | ✓ | × | × | ✓ | 0.160 |
| 51 | × | ✓ | × | × | ✓ | ✓ | × | 0.148 |
| 52 | × | ✓ | × | × | ✓ | × | ✓ | 0.163 |
| 53 | × | ✓ | × | × | × | ✓ | ✓ | 0.168 |
| 54 | × | × | ✓ | ✓ | ✓ | × | × | 0.183 |
| 55 | × | × | ✓ | ✓ | × | ✓ | × | 0.187 |
| 56 | × | × | ✓ | ✓ | × | × | ✓ | 0.159 |
| 57 | × | × | ✓ | × | ✓ | ✓ | × | 0.148 |
| 58 | × | × | ✓ | × | ✓ | × | ✓ | 0.159 |
| 59 | × | × | ✓ | × | × | ✓ | ✓ | 0.168 |
| 60 | × | × | × | ✓ | ✓ | ✓ | × | 0.149 |
| 61 | × | × | × | ✓ | ✓ | × | ✓ | 0.155 |
| 62 | × | × | × | ✓ | × | ✓ | ✓ | 0.162 |
| 63 | × | × | × | × | ✓ | ✓ | ✓ | 0.154 |
| 64 | ✓ | ✓ | ✓ | ✓ | × | × | × | 0.200 |
| 65 | ✓ | ✓ | ✓ | × | ✓ | × | × | 0.193 |
| 66 | ✓ | ✓ | ✓ | × | × | ✓ | × | 0.191 |
| 67 | ✓ | ✓ | ✓ | × | × | × | ✓ | 0.167 |
| 68 | ✓ | ✓ | × | ✓ | ✓ | × | × | 0.198 |
| 69 | ✓ | ✓ | × | ✓ | × | ✓ | × | 0.200 |
| 70 | ✓ | ✓ | × | ✓ | × | × | ✓ | 0.161 |
| 71 | ✓ | ✓ | × | × | ✓ | ✓ | × | 0.151 |
| 72 | ✓ | ✓ | × | × | ✓ | × | ✓ | 0.156 |
| 73 | ✓ | ✓ | × | × | × | ✓ | ✓ | 0.163 |
| 74 | ✓ | × | ✓ | ✓ | ✓ | × | × | 0.198 |
| 75 | ✓ | × | ✓ | ✓ | × | ✓ | × | 0.198 |
| 76 | ✓ | × | ✓ | ✓ | × | × | ✓ | 0.159 |
| 77 | ✓ | × | ✓ | × | ✓ | ✓ | × | 0.154 |
| 78 | ✓ | × | ✓ | × | ✓ | × | ✓ | 0.154 |
| 79 | ✓ | × | ✓ | × | × | ✓ | ✓ | 0.162 |
| 80 | ✓ | × | × | ✓ | ✓ | ✓ | × | 0.145 |
| 81 | ✓ | × | × | ✓ | ✓ | × | ✓ | 0.158 |
| 82 | ✓ | × | × | ✓ | × | ✓ | ✓ | 0.164 |
| 83 | ✓ | × | × | × | ✓ | ✓ | ✓ | 0.149 |
| 84 | × | ✓ | ✓ | ✓ | ✓ | × | × | 0.186 |
| 85 | × | ✓ | ✓ | ✓ | × | ✓ | × | 0.193 |
| 86 | × | ✓ | ✓ | ✓ | × | × | ✓ | 0.169 |
| 87 | × | ✓ | ✓ | × | ✓ | ✓ | × | 0.153 |
| 88 | × | ✓ | ✓ | × | ✓ | × | ✓ | 0.173 |
| 89 | × | ✓ | ✓ | × | × | ✓ | ✓ | 0.177 |
| 90 | × | ✓ | × | ✓ | ✓ | ✓ | × | 0.156 |
| 91 | × | ✓ | × | ✓ | ✓ | × | ✓ | 0.169 |
| 92 | × | ✓ | × | ✓ | × | ✓ | ✓ | 0.174 |
| 93 | × | ✓ | × | × | ✓ | ✓ | ✓ | 0.156 |
| 94 | × | × | ✓ | ✓ | ✓ | ✓ | × | 0.159 |
| 95 | × | × | ✓ | ✓ | ✓ | × | ✓ | 0.166 |
| 96 | × | × | ✓ | ✓ | × | ✓ | ✓ | 0.169 |
| 97 | × | × | ✓ | × | ✓ | ✓ | ✓ | 0.159 |
| 98 | × | × | × | ✓ | ✓ | ✓ | ✓ | 0.150 |
| 99 | ✓ | ✓ | ✓ | ✓ | ✓ | × | × | 0.203 |
| 100 | ✓ | ✓ | ✓ | ✓ | × | ✓ | × | 0.199 |
| 101 | ✓ | ✓ | ✓ | ✓ | × | × | ✓ | 0.165 |
| 102 | ✓ | ✓ | ✓ | × | ✓ | ✓ | × | 0.154 |
| 103 | ✓ | ✓ | ✓ | × | ✓ | × | ✓ | 0.164 |
| 104 | ✓ | ✓ | ✓ | × | × | ✓ | ✓ | 0.165 |
| 105 | ✓ | ✓ | × | ✓ | ✓ | ✓ | × | 0.161 |
| 106 | ✓ | ✓ | × | ✓ | ✓ | × | ✓ | 0.164 |
| 107 | ✓ | ✓ | × | ✓ | × | ✓ | ✓ | 0.171 |
| 108 | ✓ | ✓ | × | × | ✓ | ✓ | ✓ | 0.164 |
| 109 | ✓ | × | ✓ | ✓ | ✓ | ✓ | × | 0.160 |
| 110 | ✓ | × | ✓ | ✓ | ✓ | × | ✓ | 0.165 |
| 111 | ✓ | × | ✓ | ✓ | × | ✓ | ✓ | 0.169 |
| 112 | ✓ | × | ✓ | × | ✓ | ✓ | ✓ | 0.163 |
| 113 | ✓ | × | × | ✓ | ✓ | ✓ | ✓ | 0.154 |
| 114 | × | ✓ | ✓ | ✓ | ✓ | ✓ | × | 0.163 |
| 115 | × | ✓ | ✓ | ✓ | ✓ | × | ✓ | 0.175 |
| 116 | × | ✓ | ✓ | ✓ | × | ✓ | ✓ | 0.178 |
| 117 | × | ✓ | ✓ | × | ✓ | ✓ | ✓ | 0.159 |
| 118 | × | ✓ | × | ✓ | ✓ | ✓ | ✓ | 0.164 |
| 119 | × | × | ✓ | ✓ | ✓ | ✓ | ✓ | 0.162 |
| 120 | ✓ | ✓ | ✓ | ✓ | ✓ | ✓ | × | 0.161 |
| 121 | ✓ | ✓ | ✓ | ✓ | ✓ | × | ✓ | 0.171 |
| 122 | ✓ | ✓ | ✓ | ✓ | × | ✓ | ✓ | 0.173 |
| 123 | ✓ | ✓ | ✓ | × | ✓ | ✓ | ✓ | 0.162 |
| 124 | ✓ | ✓ | × | ✓ | ✓ | ✓ | ✓ | 0.164 |
| 125 | ✓ | × | ✓ | ✓ | ✓ | ✓ | ✓ | 0.165 |
| 126 | × | ✓ | ✓ | ✓ | ✓ | ✓ | ✓ | 0.166 |
| 127 | ✓ | ✓ | ✓ | ✓ | ✓ | ✓ | ✓ | 0.168 |

Table 10: **LOVM Model Selection Ablation.** Here, we ablate all the different scores used in our baselines for model ranking by mean per-class recall. We separated the text classification (C) base scores, and the granularity-based scores that quantify inter- and intra-class similarity. $a_{IN}$ - Imagenet Accuracy, $\phi_{fisher}$ - Fisher criterion, text-f1 - caption dataset f1-score, text-acc1 - text top-1 accuracy, $\gamma_{syn}$ - Synonym Consistency score, $\varphi_{sil}$ - Silhouette score, $\rho_{disp}$ - Class Dispersion score.

Model Ranking

| Row ID | $a_{IN}$ | text-f1 | text-acc1 | $\gamma_{syn}$ | $\rho_{disp}$ | $\varphi_{sil}$ | $\phi_{fisher}$ | $\tau$ (↑) | $R_5$ (↑) |
|---|---|---|---|---|---|---|---|---|---|
| 1 | ✓ | × | × | × | × | × | × | 0.186 | 0.504 |
| 2 | × | ✓ | × | × | × | × | × | 0.058 | 0.252 |
| 3 | × | × | ✓ | × | × | × | × | 0.029 | 0.191 |
| 4 | × | × | × | ✓ | × | × | × | -0.014 | 0.217 |
| 5 | × | × | × | × | ✓ | × | × | 0.014 | 0.261 |
| 6 | × | × | × | × | × | ✓ | × | -0.014 | 0.270 |
| 7 | × | × | × | × | × | × | ✓ | 0.000 | 0.157 |
| 8 | ✓ | ✓ | × | × | × | × | × | 0.200 | 0.513 |
| 9 | ✓ | × | ✓ | × | × | × | × | 0.200 | 0.513 |
| 10 | ✓ | × | × | ✓ | × | × | × | 0.116 | 0.487 |
| 11 | ✓ | × | × | × | ✓ | × | × | 0.177 | 0.530 |
| 12 | ✓ | × | × | × | × | ✓ | × | 0.186 | 0.504 |
| 13 | ✓ | × | × | × | × | × | ✓ | 0.072 | 0.478 |
| 14 | × | ✓ | ✓ | × | × | × | × | 0.029 | 0.235 |
| 15 | × | ✓ | × | ✓ | × | × | × | 0.072 | 0.243 |
| 16 | × | ✓ | × | × | ✓ | × | × | 0.014 | 0.243 |
| 17 | × | ✓ | × | × | × | ✓ | × | 0.014 | 0.252 |
| 18 | × | ✓ | × | × | × | × | ✓ | 0.014 | 0.235 |
| 19 | × | × | ✓ | ✓ | × | × | × | 0.072 | 0.252 |
| 20 | × | × | ✓ | × | ✓ | × | × | 0.043 | 0.243 |
| 21 | × | × | ✓ | × | × | ✓ | × | 0.043 | 0.226 |
| 22 | × | × | ✓ | × | × | × | ✓ | 0.043 | 0.217 |
| 23 | × | × | × | ✓ | ✓ | × | × | -0.014 | 0.217 |
| 24 | × | × | × | ✓ | × | ✓ | × | 0.000 | 0.252 |
| 25 | × | × | × | ✓ | × | × | ✓ | 0.000 | 0.174 |
| 26 | × | × | × | × | ✓ | ✓ | × | 0.000 | 0.226 |
| 27 | × | × | × | × | ✓ | × | ✓ | 0.000 | 0.165 |
| 28 | × | × | × | × | × | ✓ | ✓ | 0.000 | 0.217 |
| 29 | ✓ | ✓ | ✓ | × | × | × | × | 0.145 | 0.487 |
| 30 | ✓ | ✓ | × | ✓ | × | × | × | 0.214 | 0.504 |
| 31 | ✓ | ✓ | × | × | ✓ | × | × | 0.200 | 0.504 |
| 32 | ✓ | ✓ | × | × | × | ✓ | × | 0.180 | 0.496 |
| 33 | ✓ | ✓ | × | × | × | × | ✓ | 0.122 | 0.496 |
| 34 | ✓ | × | ✓ | ✓ | × | × | × | 0.194 | 0.496 |
| 35 | ✓ | × | ✓ | × | ✓ | × | × | 0.180 | 0.504 |
| 36 | ✓ | × | ✓ | × | × | ✓ | × | 0.180 | 0.504 |
| 37 | ✓ | × | ✓ | × | × | × | ✓ | 0.122 | 0.496 |
| 38 | ✓ | × | × | ✓ | ✓ | × | × | 0.197 | 0.548 |
| 39 | ✓ | × | × | ✓ | × | ✓ | × | 0.130 | 0.487 |
| 40 | ✓ | × | × | ✓ | × | × | ✓ | 0.072 | 0.470 |
| 41 | ✓ | × | × | × | ✓ | ✓ | × | 0.159 | 0.487 |
| 42 | ✓ | × | × | × | ✓ | × | ✓ | 0.058 | 0.487 |
| 43 | ✓ | × | × | × | × | ✓ | ✓ | 0.072 | 0.487 |
| 44 | × | ✓ | ✓ | ✓ | × | × | × | 0.000 | 0.243 |
| 45 | × | ✓ | ✓ | × | ✓ | × | × | 0.043 | 0.235 |
| 46 | × | ✓ | ✓ | × | × | ✓ | × | 0.043 | 0.235 |
| 47 | × | ✓ | ✓ | × | × | × | ✓ | 0.000 | 0.226 |
| 48 | × | ✓ | × | ✓ | ✓ | × | × | 0.029 | 0.252 |
| 49 | × | ✓ | × | ✓ | × | ✓ | × | 0.043 | 0.243 |
| 50 | × | ✓ | × | ✓ | × | × | ✓ | 0.101 | 0.252 |
| 51 | × | ✓ | × | × | ✓ | ✓ | × | 0.043 | 0.252 |
| 52 | × | ✓ | × | × | ✓ | × | ✓ | 0.043 | 0.243 |
| 53 | × | ✓ | × | × | × | ✓ | ✓ | 0.043 | 0.243 |
| 54 | × | × | ✓ | ✓ | ✓ | × | × | 0.043 | 0.243 |
| 55 | × | × | ✓ | ✓ | × | ✓ | × | 0.043 | 0.252 |
| 56 | × | × | ✓ | ✓ | × | × | ✓ | 0.101 | 0.261 |
| 57 | × | × | ✓ | × | ✓ | ✓ | × | 0.043 | 0.226 |
| 58 | × | × | ✓ | × | ✓ | × | ✓ | 0.000 | 0.226 |
| 59 | × | × | ✓ | × | × | ✓ | ✓ | 0.000 | 0.226 |
| 60 | × | × | × | ✓ | ✓ | ✓ | × | 0.000 | 0.226 |
| 61 | × | × | × | ✓ | ✓ | × | ✓ | 0.000 | 0.165 |
| 62 | × | × | × | ✓ | × | ✓ | ✓ | 0.000 | 0.209 |
| 63 | × | × | × | × | ✓ | ✓ | ✓ | 0.014 | 0.226 |
| 64 | ✓ | ✓ | ✓ | ✓ | × | × | × | 0.145 | 0.478 |
| 65 | ✓ | ✓ | ✓ | × | ✓ | × | × | 0.130 | 0.478 |
| 66 | ✓ | ✓ | ✓ | × | × | ✓ | × | 0.130 | 0.487 |
| 67 | ✓ | ✓ | ✓ | × | × | × | ✓ | 0.145 | 0.487 |
| 68 | ✓ | ✓ | × | ✓ | ✓ | × | × | 0.194 | 0.496 |
| 69 | ✓ | ✓ | × | ✓ | × | ✓ | × | 0.186 | 0.504 |
| 70 | ✓ | ✓ | × | ✓ | × | × | ✓ | 0.122 | 0.496 |
| 71 | ✓ | ✓ | × | × | ✓ | ✓ | × | 0.171 | 0.504 |
| 72 | ✓ | ✓ | × | × | ✓ | × | ✓ | 0.122 | 0.496 |
| 73 | ✓ | ✓ | × | × | × | ✓ | ✓ | 0.122 | 0.496 |
| 74 | ✓ | × | ✓ | ✓ | ✓ | × | × | 0.194 | 0.496 |
| 75 | ✓ | × | ✓ | ✓ | × | ✓ | × | 0.180 | 0.496 |
| 76 | ✓ | × | ✓ | ✓ | × | × | ✓ | 0.122 | 0.496 |
| 77 | ✓ | × | ✓ | × | ✓ | ✓ | × | 0.171 | 0.513 |
| 78 | ✓ | × | ✓ | × | ✓ | × | ✓ | 0.122 | 0.496 |
| 79 | ✓ | × | ✓ | × | × | ✓ | ✓ | 0.122 | 0.496 |
| 80 | ✓ | × | × | ✓ | ✓ | ✓ | × | 0.116 | 0.487 |
| 81 | ✓ | × | × | ✓ | ✓ | × | ✓ | 0.058 | 0.478 |
| 82 | ✓ | × | × | ✓ | × | ✓ | ✓ | 0.072 | 0.478 |
| 83 | ✓ | × | × | × | ✓ | ✓ | ✓ | 0.072 | 0.487 |
| 84 | × | ✓ | ✓ | ✓ | ✓ | × | × | 0.000 | 0.243 |
| 85 | × | ✓ | ✓ | ✓ | × | ✓ | × | 0.000 | 0.252 |
| 86 | × | ✓ | ✓ | ✓ | × | × | ✓ | 0.043 | 0.252 |
| 87 | × | ✓ | ✓ | × | ✓ | ✓ | × | 0.014 | 0.235 |
| 88 | × | ✓ | ✓ | × | ✓ | × | ✓ | 0.043 | 0.235 |
| 89 | × | ✓ | ✓ | × | × | ✓ | ✓ | 0.043 | 0.235 |
| 90 | × | ✓ | × | ✓ | ✓ | ✓ | × | 0.043 | 0.243 |
| 91 | × | ✓ | × | ✓ | ✓ | × | ✓ | 0.029 | 0.261 |
| 92 | × | ✓ | × | ✓ | × | ✓ | ✓ | 0.043 | 0.243 |
| 93 | × | ✓ | × | × | ✓ | ✓ | ✓ | 0.043 | 0.243 |
| 94 | × | × | ✓ | ✓ | ✓ | ✓ | × | 0.043 | 0.252 |
| 95 | × | × | ✓ | ✓ | ✓ | × | ✓ | 0.043 | 0.252 |
| 96 | × | × | ✓ | ✓ | × | ✓ | ✓ | 0.043 | 0.252 |
| 97 | × | × | ✓ | × | ✓ | ✓ | ✓ | 0.043 | 0.235 |
| 98 | × | × | × | ✓ | ✓ | ✓ | ✓ | 0.014 | 0.261 |
| 99 | ✓ | ✓ | ✓ | ✓ | ✓ | × | × | 0.116 | 0.478 |
| 100 | ✓ | ✓ | ✓ | ✓ | × | ✓ | × | 0.130 | 0.487 |
| 101 | ✓ | ✓ | ✓ | ✓ | × | × | ✓ | 0.145 | 0.487 |
| 102 | ✓ | ✓ | ✓ | × | ✓ | ✓ | × | 0.130 | 0.487 |
| 103 | ✓ | ✓ | ✓ | × | ✓ | × | ✓ | 0.130 | 0.478 |
| 104 | ✓ | ✓ | ✓ | × | × | ✓ | ✓ | 0.116 | 0.496 |
| 105 | ✓ | ✓ | × | ✓ | ✓ | ✓ | × | 0.180 | 0.487 |
| 106 | ✓ | ✓ | × | ✓ | ✓ | × | ✓ | 0.122 | 0.496 |
| 107 | ✓ | ✓ | × | ✓ | × | ✓ | ✓ | 0.122 | 0.496 |
| 108 | ✓ | ✓ | × | × | ✓ | ✓ | ✓ | 0.122 | 0.496 |
| 109 | ✓ | × | ✓ | ✓ | ✓ | ✓ | × | 0.180 | 0.496 |
| 110 | ✓ | × | ✓ | ✓ | ✓ | × | ✓ | 0.136 | 0.496 |
| 111 | ✓ | × | ✓ | ✓ | × | ✓ | ✓ | 0.136 | 0.504 |
| 112 | ✓ | × | ✓ | × | ✓ | ✓ | ✓ | 0.136 | 0.496 |
| 113 | ✓ | × | × | ✓ | ✓ | ✓ | ✓ | 0.072 | 0.487 |
| 114 | × | ✓ | ✓ | ✓ | ✓ | ✓ | × | 0.000 | 0.252 |
| 115 | × | ✓ | ✓ | ✓ | ✓ | × | ✓ | 0.000 | 0.243 |
| 116 | × | ✓ | ✓ | ✓ | × | ✓ | ✓ | 0.029 | 0.252 |
| 117 | × | ✓ | ✓ | × | ✓ | ✓ | ✓ | 0.014 | 0.235 |
| 118 | × | ✓ | × | ✓ | ✓ | ✓ | ✓ | 0.043 | 0.252 |
| 119 | × | × | ✓ | ✓ | ✓ | ✓ | ✓ | 0.043 | 0.261 |
| 120 | ✓ | ✓ | ✓ | ✓ | ✓ | ✓ | × | 0.130 | 0.496 |
| 121 | ✓ | ✓ | ✓ | ✓ | ✓ | × | ✓ | 0.116 | 0.496 |
| 122 | ✓ | ✓ | ✓ | ✓ | × | ✓ | ✓ | 0.116 | 0.504 |
| 123 | ✓ | ✓ | ✓ | × | ✓ | ✓ | ✓ | 0.116 | 0.487 |
| 124 | ✓ | ✓ | × | ✓ | ✓ | ✓ | ✓ | 0.122 | 0.496 |
| 125 | ✓ | × | ✓ | ✓ | ✓ | ✓ | ✓ | 0.122 | 0.496 |
| 126 | × | ✓ | ✓ | ✓ | ✓ | ✓ | ✓ | 0.043 | 0.252 |
| 127 | ✓ | ✓ | ✓ | ✓ | ✓ | ✓ | ✓ | 0.116 | 0.487 |

ranking. The unified approach also captures more significant ranking variation in the non-natural image / non-classification benchmarks.

## C.5 Domain Shift Experiment

An obvious obstacle to text-based model performance prediction methods is the difficulty in describing distribution shifts. For example, VLMs evaluated on ImageNet and ImageNet-v2 will get the same text-predicted accuracy while the actual performance differs. Meanwhile, for some domain shifts - like ImageNet and ImageNet sketch - this shift can be described via text. We want to evaluate how capable text-only methods are at estimating the dataset difficulty under such shifts and compare them to well-accepted image-based approaches.

**Dataset Description Similarity.** We extract each dataset's description from either the abstract or introduction section of the original manuscript. Subsequently, we extract the text embeddings for all dataset descriptions using a pre-trained CLIP model. We then compute the cosine similarity

Table 11: **LOVM Model Prediction Ablation.** Here, we ablate all the different scores used in our baselines for mean per-class recall prediction. We separated the text classification (C) base scores, and the granularity-based scores that quantify inter- and intra-class similarity. $a_{\text{IN}}$ - Imagenet Accuracy, $\phi_{\text{fisher}}$ - Fisher criterion, text-f1 - caption dataset f1-score, text-acc1 - text top-1 accuracy, $\gamma_{\text{syn}}$ - Synonym Consistency score, $\varphi_{\text{sil}}$ - Silhouette score, $\rho_{\text{disp}}$ - Class Dispersion score.

| Row ID | $a_{\text{IN}}$ | text-f1 | text-acc1 | $\gamma_{\text{syn}}$ | $\rho_{\text{disp}}$ | $\varphi_{\text{sil}}$ | $\phi_{\text{fisher}}$ | $L_1 (\downarrow)$ |
|---|---|---|---|---|---|---|---|---|
| 1 | ✓ | × | × | × | × | × | × | 0.228 |
| 2 | × | ✓ | × | × | × | × | × | 0.182 |
| 3 | × | × | ✓ | × | × | × | × | 0.183 |
| 4 | × | × | × | ✓ | × | × | × | 0.192 |
| 5 | × | × | × | × | ✓ | × | × | 0.206 |
| 6 | × | × | × | × | × | ✓ | × | 0.232 |
| 7 | × | × | × | × | × | × | ✓ | 0.175 |
| 8 | ✓ | ✓ | × | × | × | × | × | 0.196 |
| 9 | ✓ | × | ✓ | × | × | × | × | 0.190 |
| 10 | ✓ | × | × | ✓ | × | × | × | 0.226 |
| 11 | ✓ | × | × | × | ✓ | × | × | 0.219 |
| 12 | ✓ | × | × | × | × | ✓ | × | 0.229 |
| 13 | ✓ | × | × | × | × | × | ✓ | 0.178 |
| 14 | × | ✓ | ✓ | × | × | × | × | 0.182 |
| 15 | × | ✓ | × | ✓ | × | × | × | 0.192 |
| 16 | × | ✓ | × | × | ✓ | × | × | 0.194 |
| 17 | × | ✓ | × | × | × | ✓ | × | 0.189 |
| 18 | × | ✓ | × | × | × | × | ✓ | 0.165 |
| 19 | × | × | ✓ | ✓ | × | × | × | 0.190 |
| 20 | × | × | ✓ | × | ✓ | × | × | 0.190 |
| 21 | × | × | ✓ | × | × | ✓ | × | 0.188 |
| 22 | × | × | ✓ | × | × | × | ✓ | 0.162 |
| 23 | × | × | × | ✓ | ✓ | × | × | 0.212 |
| 24 | × | × | × | ✓ | × | ✓ | × | 0.223 |
| 25 | × | × | × | ✓ | × | × | ✓ | 0.160 |
| 26 | × | × | × | × | ✓ | ✓ | × | 0.152 |
| 27 | × | × | × | × | ✓ | × | ✓ | 0.180 |
| 28 | × | × | × | × | × | ✓ | ✓ | 0.181 |
| 29 | ✓ | ✓ | ✓ | × | × | × | × | 0.197 |
| 30 | ✓ | ✓ | × | ✓ | × | × | × | 0.203 |
| 31 | ✓ | ✓ | × | × | ✓ | × | × | 0.197 |
| 32 | ✓ | ✓ | × | × | × | ✓ | × | 0.194 |
| 33 | ✓ | ✓ | × | × | × | × | ✓ | 0.164 |
| 34 | ✓ | × | ✓ | ✓ | × | × | × | 0.203 |
| 35 | ✓ | × | ✓ | × | ✓ | × | × | 0.196 |
| 36 | ✓ | × | ✓ | × | × | ✓ | × | 0.193 |
| 37 | ✓ | × | ✓ | × | × | × | ✓ | 0.163 |
| 38 | ✓ | × | × | ✓ | ✓ | × | × | 0.228 |
| 39 | ✓ | × | × | ✓ | × | ✓ | × | 0.232 |
| 40 | ✓ | × | × | ✓ | × | × | ✓ | 0.165 |
| 41 | ✓ | × | × | × | ✓ | ✓ | × | 0.156 |
| 42 | ✓ | × | × | × | ✓ | × | ✓ | 0.178 |
| 43 | ✓ | × | × | × | × | ✓ | ✓ | 0.179 |
| 44 | × | ✓ | ✓ | ✓ | × | × | × | 0.192 |
| 45 | × | ✓ | ✓ | × | ✓ | × | × | 0.192 |
| 46 | × | ✓ | ✓ | × | × | ✓ | × | 0.191 |
| 47 | × | ✓ | ✓ | × | × | × | ✓ | 0.173 |
| 48 | × | ✓ | × | ✓ | ✓ | × | × | 0.197 |
| 49 | × | ✓ | × | ✓ | × | ✓ | × | 0.197 |
| 50 | × | ✓ | × | ✓ | × | × | ✓ | 0.165 |
| 51 | × | ✓ | × | × | ✓ | ✓ | × | 0.156 |
| 52 | × | ✓ | × | × | ✓ | × | ✓ | 0.171 |
| 53 | × | ✓ | × | × | × | ✓ | ✓ | 0.176 |
| 54 | × | × | ✓ | ✓ | ✓ | × | × | 0.195 |
| 55 | × | × | ✓ | ✓ | × | ✓ | × | 0.192 |
| 56 | × | × | ✓ | ✓ | × | × | ✓ | 0.160 |
| 57 | × | × | ✓ | × | ✓ | ✓ | × | 0.153 |
| 58 | × | × | ✓ | × | ✓ | × | ✓ | 0.170 |
| 59 | × | × | ✓ | × | × | ✓ | ✓ | 0.174 |
| 60 | × | × | × | ✓ | ✓ | ✓ | × | 0.141 |
| 61 | × | × | × | ✓ | ✓ | × | ✓ | 0.165 |
| 62 | × | × | × | ✓ | × | ✓ | ✓ | 0.166 |
| 63 | × | × | × | × | ✓ | ✓ | ✓ | 0.160 |
| 64 | ✓ | ✓ | ✓ | ✓ | × | × | × | 0.206 |
| 65 | ✓ | ✓ | ✓ | × | ✓ | × | × | 0.201 |
| 66 | ✓ | ✓ | ✓ | × | × | ✓ | × | 0.199 |
| 67 | ✓ | ✓ | ✓ | × | × | × | ✓ | 0.170 |
| 68 | ✓ | ✓ | × | ✓ | ✓ | × | × | 0.210 |
| 69 | ✓ | ✓ | × | ✓ | × | ✓ | × | 0.203 |
| 70 | ✓ | ✓ | × | ✓ | × | × | ✓ | 0.168 |
| 71 | ✓ | ✓ | × | × | ✓ | ✓ | × | 0.161 |
| 72 | ✓ | ✓ | × | × | ✓ | × | ✓ | 0.169 |
| 73 | ✓ | ✓ | × | × | × | ✓ | ✓ | 0.172 |
| 74 | ✓ | × | ✓ | ✓ | ✓ | × | × | 0.207 |
| 75 | ✓ | × | ✓ | ✓ | × | ✓ | × | 0.205 |
| 76 | ✓ | × | ✓ | ✓ | × | × | ✓ | 0.166 |
| 77 | ✓ | × | ✓ | × | ✓ | ✓ | × | 0.162 |
| 78 | ✓ | × | ✓ | × | ✓ | × | ✓ | 0.169 |
| 79 | ✓ | × | ✓ | × | × | ✓ | ✓ | 0.171 |
| 80 | ✓ | × | × | ✓ | ✓ | ✓ | × | 0.152 |
| 81 | ✓ | × | × | ✓ | ✓ | × | ✓ | 0.167 |
| 82 | ✓ | × | × | ✓ | × | ✓ | ✓ | 0.173 |
| 83 | ✓ | × | × | × | ✓ | ✓ | ✓ | 0.163 |
| 84 | × | ✓ | ✓ | ✓ | ✓ | × | × | 0.195 |
| 85 | × | ✓ | ✓ | ✓ | × | ✓ | × | 0.197 |
| 86 | × | ✓ | ✓ | ✓ | × | × | ✓ | 0.170 |
| 87 | × | ✓ | ✓ | × | ✓ | ✓ | × | 0.160 |
| 88 | × | ✓ | ✓ | × | ✓ | × | ✓ | 0.180 |
| 89 | × | ✓ | ✓ | × | × | ✓ | ✓ | 0.182 |
| 90 | × | ✓ | × | ✓ | ✓ | ✓ | × | 0.156 |
| 91 | × | ✓ | × | ✓ | ✓ | × | ✓ | 0.173 |
| 92 | × | ✓ | × | ✓ | × | ✓ | ✓ | 0.177 |
| 93 | × | ✓ | × | × | ✓ | ✓ | ✓ | 0.165 |
| 94 | × | × | ✓ | ✓ | ✓ | ✓ | × | 0.156 |
| 95 | × | × | ✓ | ✓ | ✓ | × | ✓ | 0.172 |
| 96 | × | × | ✓ | ✓ | × | ✓ | ✓ | 0.173 |
| 97 | × | × | ✓ | × | ✓ | ✓ | ✓ | 0.163 |
| 98 | × | × | × | ✓ | ✓ | ✓ | ✓ | 0.151 |
| 99 | ✓ | ✓ | ✓ | ✓ | ✓ | × | × | 0.211 |
| 100 | ✓ | ✓ | ✓ | ✓ | × | ✓ | × | 0.203 |
| 101 | ✓ | ✓ | ✓ | ✓ | × | × | ✓ | 0.172 |
| 102 | ✓ | ✓ | ✓ | × | ✓ | ✓ | × | 0.164 |
| 103 | ✓ | ✓ | ✓ | × | ✓ | × | ✓ | 0.176 |
| 104 | ✓ | ✓ | ✓ | × | × | ✓ | ✓ | 0.178 |
| 105 | ✓ | ✓ | × | ✓ | ✓ | ✓ | × | 0.162 |
| 106 | ✓ | ✓ | × | ✓ | ✓ | × | ✓ | 0.177 |
| 107 | ✓ | ✓ | × | ✓ | × | ✓ | ✓ | 0.178 |
| 108 | ✓ | ✓ | × | × | ✓ | ✓ | ✓ | 0.172 |
| 109 | ✓ | × | ✓ | ✓ | ✓ | ✓ | × | 0.162 |
| 110 | ✓ | × | ✓ | ✓ | ✓ | × | ✓ | 0.175 |
| 111 | ✓ | × | ✓ | ✓ | × | ✓ | ✓ | 0.177 |
| 112 | ✓ | × | ✓ | × | ✓ | ✓ | ✓ | 0.172 |
| 113 | ✓ | × | × | ✓ | ✓ | ✓ | ✓ | 0.165 |
| 114 | × | ✓ | ✓ | ✓ | ✓ | ✓ | × | 0.160 |
| 115 | × | ✓ | ✓ | ✓ | ✓ | × | ✓ | 0.178 |
| 116 | × | ✓ | ✓ | ✓ | × | ✓ | ✓ | 0.182 |
| 117 | × | ✓ | ✓ | × | ✓ | ✓ | ✓ | 0.168 |
| 118 | × | ✓ | × | ✓ | ✓ | ✓ | ✓ | 0.165 |
| 119 | × | × | ✓ | ✓ | ✓ | ✓ | ✓ | 0.164 |
| 120 | ✓ | ✓ | ✓ | ✓ | ✓ | ✓ | × | 0.164 |
| 121 | ✓ | ✓ | ✓ | ✓ | ✓ | × | ✓ | 0.181 |
| 122 | ✓ | ✓ | ✓ | ✓ | × | ✓ | ✓ | 0.182 |
| 123 | ✓ | ✓ | ✓ | × | ✓ | ✓ | ✓ | 0.173 |
| 124 | ✓ | ✓ | × | ✓ | ✓ | ✓ | ✓ | 0.172 |
| 125 | ✓ | × | ✓ | ✓ | ✓ | ✓ | ✓ | 0.171 |
| 126 | × | ✓ | ✓ | ✓ | ✓ | ✓ | ✓ | 0.168 |
| 127 | ✓ | ✓ | ✓ | ✓ | ✓ | ✓ | ✓ | 0.175 |

*(Left-hand side row label: Metric Prediction)*

between the descriptions of the downstream datasets to the original pre-training dataset to quantify how different the two datasets are.

**Prompt Embedding Similarity.** We use the cosine similarity between dataset-specific and generic class prompts to evaluate domain shift. Specifically, based on the original list of class prompts from CLIP, we pick the ones that best describe our target dataset. For instance, for the ImageNet-sketch dataset, we selected prompts such as "A sketch of a {c}" or "A doodle of a {c}". Then, we use the text encoder from a pre-trained CLIP model to extract embeddings from dataset-specific and generic class prompts and compute the cosine similarity between each pair. We use the mean cosine similarity to measure how similar the target dataset is to the pre-training dataset. The dataset-specific prompts can be found in the following subsection.

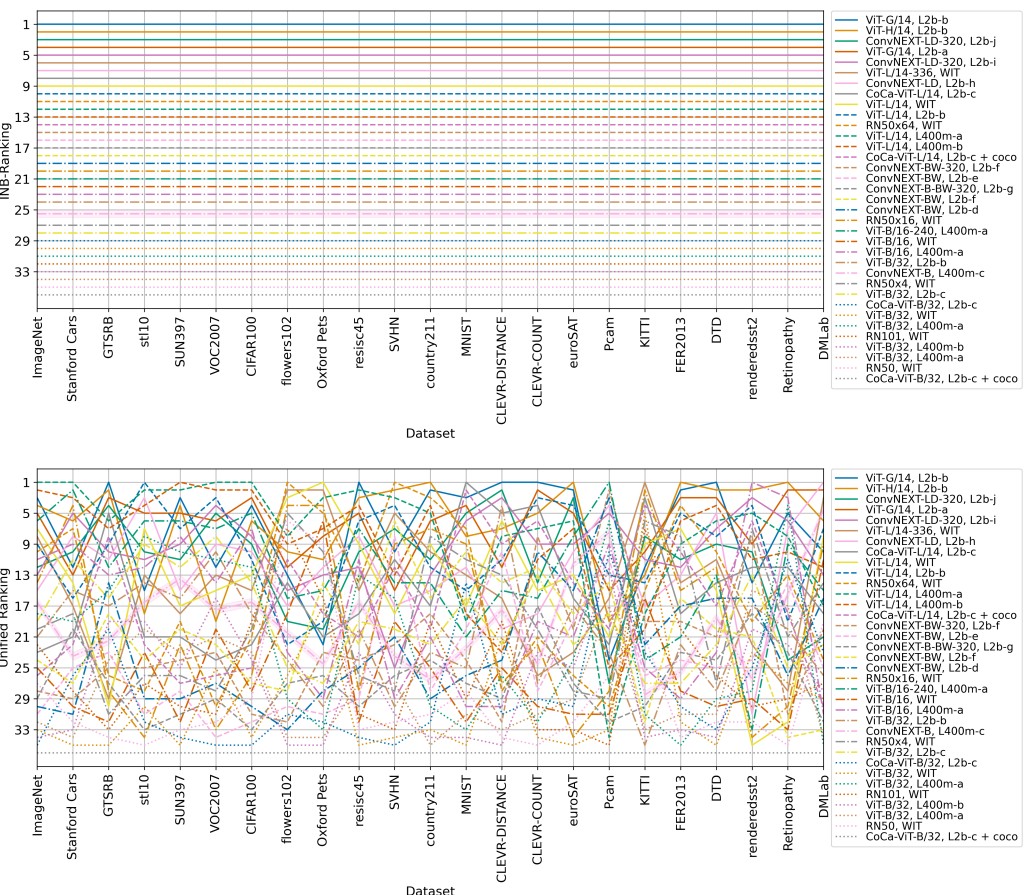

Figure 9: **Raw Model Ranking.** (top) ImageNet benchmark approach assumes the same model ranking for all datasets and cannot predict fine-grained model ranking. (bottom) The unified approach can adjust the coarse ImageNet rankings for a more realistic model ranking.

**Image-Text Embedding.** We wanted to compare with widely used dataset difficulty approaches. One common approach is to use the confidence of a model's prediction to determine a dataset's difficulty. To do so, we first $n$ images from the target dataset and extract image embeddings for each of the $n$ images. This simulates the scenario where we only have access to $n$ images from the target dataset to estimate model performance, where $n$ is much smaller than the dataset size. Then, we embed the class prompt into text embeddings and compute the prediction logits between each image embedding and class embeddings. Lastly, we compute the *entropy score* Ethayarajh et al. [2022] and *max prediction logit* Feng et al. [2022] to determine dataset difficulty.

**Image Embedding Distance.** Another common approach is estimating the difference in distribution between the test and train sets Scheidegger et al. [2021]. We, therefore, use the distance between the target and pre-training image embeddings to quantify dataset difficulty. Similar to the image-text embedding approach, we first sample $n$ images from the target dataset to extract image embedding using a pre-trained CLIP image encoder. Additionally, we sample $m$ images from the pre-training dataset to extract image embeddings. We only sample $m$ examples since these VLMs are typically pre-trained on an internet-scale dataset, which makes it challenging to embed and compute distance measures on the entire pre-training dataset. We then compute the $L_2$ distances between the target and pre-training datasets. We use *max, min & mean $L_2$* to quantify dataset difficulty.

**Datasets.** To evaluate the feasibility and effectiveness of each, we use the following variants of ImageNet. Each dataset captures a different distribution shift from the original ImageNet:

- **ImageNet**: The original ImageNet dataset.

Table 12: **Dataset difficulty prediction.** Here we evaluate different method's ability to rank variations of dataset based on their difficulty. Ground truth is based on CLIP's zero-shot performance on each dataset. We evaluate each method based on Kendall's rank correlation ($\tau$). IN - ImageNet, V2 - ImageNet-v2, A - ImageNet Adversarial, R - Imagenet Rendition, S - ImageNet Sketch.

| Method | Metric | Raw Value | | | | | Rank | | | | | $\tau$ (↑) |
|---|---|---|---|---|---|---|---|---|---|---|---|---|
| | | IN | V2 | A | R | S | IN | V2 | A | R | S | |
| True Performance | acc (↑) | 0.762 | 0.701 | 0.771 | 0.889 | 0.602 | 3 | 4 | 2 | 1 | 5 | 1.000 |
| Text Sim. | cosine (↑) | 0.807 | 0.808 | 0.781 | 0.832 | 0.786 | 3 | 2 | 5 | 1 | 4 | 0.200 |
| Prompt Embedding Sim. | cosine (↑) | 0.820 | 0.820 | 0.801 | 0.808 | 0.774 | 1 | 1 | 4 | 3 | 5 | 0.105 |
| Image-Text Embedding | entropy (↑) | 10.819 ± 1.6e-4 | 9.210 ± 1.5e-4 | 8.922 ± 1.8e-4 | 10.309 ± 1.4e-4 | 10.837 ± 1.4e-4 | 2 | 5 | 4 | 3 | 1 | -0.200 |
| Image-Text Embedding | max logits (↑) | 0.273 ± 0.032 | 0.264 ± 0.035 | 0.252 ± 0.027 | 0.260 ± 0.028 | 0.271 ± 0.029 | 1 | 3 | 5 | 4 | 2 | -0.400 |
| Image Embedding Dist. | min $L_2$ (↓) | 0.768 ± 0.084 | 0.790 ± 0.090 | 0.850 ± 0.068 | 0.796 ± 0.079 | 0.753 ± 0.093 | 2 | 3 | 5 | 4 | 1 | -0.600 |
| Image Embedding Dist. | mean $L_2$ (↓) | 1.422 ± 0.013 | 1.414 ±0.014 | 1.412 ± 0.013 | 0.413 ± 0.014 | 1.411 ± 0.012 | 5 | 4 | 2 | 3 | 1 | -0.200 |
| Image Embedding Dist. | max $L_2$ (↓) | 1.099 ± 0.036 | 1.103 ± 0.041 | 1.131 ± 0.033 | 1.089 ± 0.038 | 1.077 ± 0.033 | 3 | 4 | 5 | 2 | 1 | -0.200 |

- **ImageNet Version 2 (ImageNet-v2)**: A new test set for ImageNet sampled a decade later.
- **ImageNet Sketch (ImageNet-s)**: A ImageNet test set dataset with sketch-like iamges.
- **ImageNet Rendition (ImageNet-r)**: contains art, cartoons, deviantart, graffiti, embroidery, graphics, origami, paintings, patterns, plastic objects, plush objects, sculptures, sketches, tattoos, toys, and video game renditions of ImageNet classes.
- **ImageNet Adversarial (ImageNet-a)**: A real-world distribution shift ImageNet dataset with changes in image style, blurriness, camera operation, and geographic location.

We extract descriptions of each dataset from either the abstract or induction section of their original manuscript. The description used for each dataset is as shown here:

- **LAION400m**: *"a dataset with CLIP-filtered 400 million image-text pairs."*
- **ImageNet**: *"a benchmark in object category classification and detection on hundreds of object categories."*
- **ImageNet Version 2**: *"three test sets with 10,000 new images each. Importantly, these test sets were sampled after a decade of progress on the original ImageNet dataset."*
- **ImageNet Adversarial**: *"real-world distribution shift datasets consisting of changes in image style, image blurriness, geographic locations."*
- **ImageNet Rendition**: *"art, cartoons, DeviantArt, graffiti, embroidery, graphics, origami, paintings, patterns, plastic objects, plush objects, sculptures, sketches, tattoos, toys, and video game renditions of ImageNet classes."*
- **ImageNet Sketch**: *"a new dataset consisting of sketch-like images, that matches the ImageNet classification validation set in categories and scale"*

The dataset-specific prompts used for the prompt embedding distance metric are listed in Fig. 10.

**Evaluation.** We use Kendall's rank correlation ($\tau$) to evaluate our method's ability to rank the datasets in terms of their difficulties. Since *image-text embedding* and *ImageNet embedding distance* require sampling from the target dataset, we run our evaluation 1,000 times with different samples and compute the average metric. We also compute the standard deviations of the 1,000 run to estimate the variability of random samples.

We show the results of using our strategies to estimate domain shift in Tab 12. Based on the results, it is clear that none of the current methods can capture the dataset difficulty. Furthermore, the variability based on the standard deviation makes our results heavily dependent on the samples drawn from the target dataset, again suggesting these approaches' limitations.

# D Limitations

Our study, while extensive, is not without limitations. Primarily, our focus rests on zero-shot tasks due to the nature of the LOVM's design. The framework's primary aim is to determine the best model for a given task when there is **no** access to the downstream task dataset. Under these circumstances, fine-tuning or linear probing is not viable, as they require access to labeled or unlabeled images from

```python
# imagenet prompts
imagenet = [
f'a bad photo of a {c}.',
f'a photo of many {c}.',
f'a bright photo of a {c}.',
f'a photo of a clean {c}.',
f'a photo of a dirty {c}.',
f'a photo of my {c}.',
f'a photo of the cool {c}.',
f'a close-up photo of a {c}.',
f'a bright photo of the {c}.',
f'a photo of the dirty {c}.',
f'a photo of the {c}.',
f'a good photo of the {c}.',
f'a photo of one {c}.',
f'a close-up photo of the {c}.',
f'a photo of a {c}.',
f'a photo of the clean {c}.',
f'a photo of a large {c}.',
f'a photo of a nice {c}.',
f'a good photo of a {c}.',
f'a photo of the nice {c}.',
f'a photo of the small {c}.',
f'a photo of the weird {c}.',
f'a photo of the large {c}.',
f'a photo of a cool {c}.',
f'a photo of a small {c}.',
]

# imagenet v2 prompts
imagenet_v2 = [
f'a bad photo of a {c}.',
f'a photo of many {c}.',
f'a bright photo of a {c}.',
f'a photo of a clean {c}.',
f'a photo of a dirty {c}.',
f'a photo of my {c}.',
f'a photo of the cool {c}.',
f'a close-up photo of a {c}.',
f'a bright photo of the {c}.',
f'a photo of the dirty {c}.',
f'a photo of the {c}.',
f'a good photo of the {c}.',
f'a photo of one {c}.',
f'a close-up photo of the {c}.',
f'a photo of a {c}.',
f'a photo of the clean {c}.',
f'a photo of a large {c}.',
f'a photo of a nice {c}.',
f'a good photo of a {c}.',
f'a photo of the nice {c}.',
f'a photo of the small {c}.',
f'a photo of the weird {c}.',
f'a photo of the large {c}.',
f'a photo of a cool {c}.',
f'a photo of a small {c}.',
]
```

```python
# imagenet-a prompts
imagenet-a = [
f'a bad photo of a {c}.',
f'a bad photo of the {c}.',
f'a cropped photo of the {c}.',
f'a photo of a hard to see {c}.',
f'a photo of a dirty {c}.',
f'a dark photo of the {c}.',
f'a pixelated photo of the {c}.',
f'a cropped photo of a {c}.',
f'a photo of the dirty {c}.',
f'a blurry photo of the {c}.',
f'a photo of a weird {c}.',
f'a blurry photo of a {c}.',
f'a pixelated photo of a {c}.',
f'a photo of the weird {c}.',
]

# imagenet-s prompts
imagenet-s = [
f'a drawing of a {c}.',
f'a doodle of a {c}.',
f'a sketch of a {c}.',
f'a doodle of the {c}.',
f'a sketch of the {c}.',
]

# imagenet-r prompts
imagenet-r = [
f'a sculpture of a {c}.',
f'a rendering of a {c}.',
f'graffiti of a {c}.',
f'a tattoo of a {c}.',
f'the embroidered {c}.',
f'a drawing of a {c}.',
f'the plastic {c}.',
f'a painting of the {c}.',
f'a painting of a {c}.',
f'a sculpture of the {c}.',
f'a plastic {c}.',
f'a rendering of the {c}.',
f'a {c} in a video game.',
f'the origami {c}.',
f'the {c} in a video game.',
f'a origami {c}.',
f'the toy {c}.',
f'a rendition of a {c}.',
f'a cartoon {c}.',
f'art of a {c}.',
f'a sketch of the {c}.',
f'a embroidered {c}.',
f'a plushie {c}.',
f'the cartoon {c}.',
f'the plushie {c}.',
f'graffiti of the {c}.',
f'a toy {c}.',
f'a tattoo of the {c}.'
]
```

Figure 10: **Prompting Templates Examples.** Above can be examples of different prompting templates used in the study. When using a prompting template, the '{c}' character is replaced by the class name.

the downstream task dataset. If such data were available, the more straightforward approach would be to address the conventional transferability problem as detailed in prior works. The ideal scenario we envision for using LOVM is one where a user with minimal technical expertise seeks to conduct a vision task. In this situation, the user can utilize a LOVM method to discern the most suitable model and the relevant classes, enabling them to deploy the model without needing to delve into technical nuances. However, if one possesses data for fine-tuning, conducting a direct evaluation on this small dataset is likely the most accurate course of action. This constraint stems from the fact that LOVM methods cannot make differential predictions without access to the fine-tuning data. Predicting the performance after fine-tuning or linear probing would heavily depend on the correlation between the results pre and post- fine-tuning/linear probing, a scenario we aim to avoid in the design of LOVM. However, previous work has shown some correlation exists, so there may be some transferability to fine-tuned/linear probed models [Wong et al., 2022].

Secondly, as discussed in Sec. C.5, even datasets bearing identical content may encounter a domain shift. Such shifts can be clearly explained in some cases, such as when comparing ImageNet-regular/rendition/sketch, but in others, the shift may be more elusive. For instance, when comparing ImageNet to ImageNet-a, or when class distribution shifts occur, identifying the source of the shift becomes challenging. In these scenarios, LOVM methods might struggle to accurately predict the performance of a VLM, though model selection might be marginally affected.

Thirdly, the scope of VLMs in our work is currently confined to those trained with a contrastive loss. The contrastive loss is central to the cross-modal transferability [Liang et al., 2022, Zhang et al., 2023, Eyuboglu et al., 2022], and it is currently unclear if models not utilizing any contrastive loss will exhibit the same behavior. Additional architectures, such as ones using unified text and image encoders, is an interesting research direction and can also be incorporated in future works.

Finally, while the utility of text-only solutions described in Sec. C.5 warrants continued investigation, it may be necessary to incorporate unlabeled test images to gauge domain shifts. Combining LOVM methods with these image-based evaluations remains a promising area of ongoing research.

# E  Broader Impacts

Our work simplifies selecting vision-language models (VLMs) for specific tasks, increasing the accessibility of artificial intelligence (AI) applications. However, this accessibility may be a double-edged sword. On the one hand, it could democratize AI applications, allowing smaller entities or independent researchers to utilize AI technologies more effectively. On the other hand, this easy access might also enable malicious entities to deploy harmful applications more readily, posing risks to sectors such as information security and personal privacy.

Moreover, despite our methodology's efficiencies, it carries the risk of sub-optimal model selection due to inherent limitations. Inaccuracies could lead to inefficient resource allocation or inferior performance in real-world applications, particularly in high-stakes fields such as healthcare or autonomous driving. Overall, while our work contributes to the efficiency and accessibility of AI applications, it highlights the need for vigilance and continuous refinement to mitigate potential negative impacts.