# OpenReview forum: "LOVM: Language-Only Vision Model Selection"
_NeurIPS.cc/2023/Track/Datasets_and_Benchmarks — NeurIPS 2023 Datasets and Benchmarks Poster_

### Official Review · Reviewer_BH74 · 2023-07-15
**A novel approach to select VLMs**

**Rating:** 7
**Confidence:** 4
**Correctness:** Mostly correct. See weaknesses/limita…
**Clarity:** It is overall well-written.

**Strengths:**

- This paper introduces a novel and practically useful problem setting - selecting vision models using only text descriptions of the target task. This eliminates the need for collecting and curating image datasets. LOVM can also give predictions of the performance of the downstream task.
- The experiments are extensive - evaluation with 35 pre-trained VLMs on 23 datasets. The authors also use various metrics for evaluations.
- This paper empirically demonstrated the effectiveness of cross-modality transferability for VLM performance estimation and provides some insightful analysis of VLMs behavior based on the experimental results.



**Additional Feedback:**

NA

**Documentation:**

The code is open-sourced and the authors have provided detailed documentation in the appendix.

**Ethics:**

No ethical concerns.

**Limitations:**

The datasets used in the benchmark are mostly classification datasets and the datasets for other tasks have fewer classes than classification tasks. Would that imbalance somehow affect the conclusions for other tasks?

**Opportunities For Improvement:**

- The authors utilized VLMs in the open-clip library. While there're many different backbones and pretrained datasets. They are mostly pretrained in the CLIP or CoCa style. The VLMs pretrained in other objectives are missing and thus it is unknown if the proposed LOVM can generalize to other methods.
- The proposed LOVM can be only applied to VLMs with modality-specific encoders (e.g., CLIP), but not suitable for the unified architecture where different modalities share the same backbone. This may limit the practicability as unified architecture is becoming more and more popular.

**Relation To Prior Work:**

Previous works are appropriately discussed.

**Summary And Contributions:**

This paper proposes a new task called Language-Only Vision Model (LOVM) selection, where the goal is to select the best-performing vision-language model for a given downstream visual zero-shot application using only a textual description of that task. They create a benchmark dataset and baselines using text generation and cross-modality transferability and show the feasibility of model selection from text alone, outperforming image-based selection.

---

> ### Author Response · Authors · 2023-08-08
>
> Dear R-BH74,
>
> Thank you for the detailed feedback and for recognizing the strengths and contributions of our work. We address the opportunities for improvement and concerns raised:
>
> **1. Limited Scope of Pretrained VLMs:**
>
> Our work focuses on the evaluation of VLMs trained with contrastive losses, as they were shown to exhibit the phenomenon of cross-modality transfer (i.e. text-based predictions are good proxies for image-based predictions) as rigorously investigated by previous works [Liang et al. neurIPS 2022, Zhang et. al. ICLR 2023, Eyuboglu et al. ICLR 2022, Gu et. al, arXiv 2022, Jiang et. al. CVPR 2023]. These studies did not consider VLMs trained without contrastive loss.  One reason is that contrastive loss is used in the majority of VLMs.  The other is that it strictly enforces the similarity and orthogonality of text and image embeddings virtually in its training objective. Whether or not LOVM is applicable in VLMs not utilizing a contrastive loss remains to be an open question that is beyond the scope of this work. As the cross-modal transferability field continues to grow to encompass more VLMs, LOVM can naturally benefit from the increased coverage. We thank you for your insight and have incorporated this discussion into our revision’s limitations (Sec. D of the appendix).
>
>
> **2. Applicability to Unified Architectures:**
>
> Your point is valid. Our work here aims to introduce the task of LOVM as a whole, and we have selected the most broadly used VLM currently in use.  However, it's worth mentioning that the task’s applicability hinges less on the architecture's type (modality-specific or unified) and more on the training objective used. If unified architectures employ a contrastive loss akin to CLIP, they would likely exhibit cross-modal transferability and could therefore be incorporated into LOVM. Exploring whether unified architectures could also be incorporated into a LOVM benchmark is a valuable research direction in future work.
>
>
> **3. The datasets used in the benchmark are mostly classification datasets and the datasets for other tasks have fewer classes than classification tasks. Would that imbalance somehow affect the conclusions for other tasks?**
>
> As part of the appendix, we have included a per-dataset breakdown of our LOVM baselines on each dataset. As similar trends can be seen for both classification and non-classification (OCR, geolocalization, distance est., etc.), we believe that the conclusion that LOVM is possible also for non-classification tasks holds as well.
>
>
> Once again, we're grateful for your constructive critique, which will guide our revisions and subsequent research endeavors.
>
> Best regards,
>
> The Authors

---

> > ### Author Response · Authors · 2023-08-24
> >
> > Dear R-BH74,
> >
> > With the review period concluding in five days, we would like to take this opportunity to emphasize our availability for any additional questions or clarifications you may have regarding our work “LOVM: Language-Only Vision Model selection”. Your insights and inquiries have been invaluable thus far, and we are committed to addressing any further concerns you may have in a timely manner.
> >
> > Best regards,
> >
> > The Authors

---

### Official Review · Reviewer_nEyK · 2023-07-20
**an interesting paper**

**Rating:** 7
**Confidence:** 3
**Clarity:** Yes.

**Strengths:**

1. The problem is overall interesting.

2. The code is open source and seems easy to use.

3. The benchmark covers a diverse collection of pre-trained VLMs spanning various architectures and encompasses a comprehensive range of tasks, and the datasets span diverse domains.

4. There is an established baseline for the benchmark which will foster continued improvement.

**Additional Feedback:**

The paper tackles an overall interesting problem. A more thorough discussion of the motivation could elevate it.

**Correctness:**

The claims are correct, I think the downstream task types of this benchmark should be increased, and the selected data set should be more suitable for the zero-shot setting.

**Documentation:**

Yes, they have provided the code for this benchmark.

**Ethics:**

No ethical concerns.

**Limitations:**

There are no potential negative societal impacts of their work

**Opportunities For Improvement:**

1. Most of the selected tasks are classification tasks, why there are no segmentation and detection tasks.

2. In actual scenarios, if we have a large-scale data set like Imagenet, we usually choose to fine-tune VLM instead of direct zero-shot to use it. So the motivation for this article has some flaws. I think that downstream tasks should choose small-scale datasets suitable for zero-shot learning.

3. I noticed that the CSV file in the supplementary material gives the value of Ground Truth. However, the difference between these ground truth accuracy rates is very small, and some of them only differ by 0.001, so GT may fluctuate due to changes in the training environment. But the effect of this fluctuation will cause strong changes in Kendall's coefficient.

**Relation To Prior Work:**

Yes.

**Summary And Contributions:**

1. The introduction of the LOVM task and benchmark for evaluating pre-trained VLMs without access to the downstream task dataset.

2. The proposal of a set of baseline models and scores for evaluating the performance of pre-trained VLMs on the LOVM benchmark.

3. An empirical evaluation of the proposed baseline models and scores on a set of pre-trained VLMs, demonstrating their effectiveness in predicting the performance of VLMs on downstream tasks.

---

> ### Author Response · Authors · 2023-08-08
>
> Dear R-nEyK,
>
> Thank you for your comprehensive review and valuable insights. We appreciate the detailed feedback, and we are addressing each of your points as follows:
>
> **1. Inclusion of Segmentation and Detection Tasks:**
>
> The proposed LOVM benchmark consists of 9 natural image classification datasets, 6 non-natural image classification tasks, and 8 non-classification tasks. Unlike detection and segmentation, all these tasks could be performed by the same VLMs and therefore allowed the comparison of LOVM methods. They also allowed the use of open_clip, a unified library encompassing many VLMs, reducing the overhead of implementing multiple methods.
>
> It is worth noting that previous transferability works have shown that these approaches may generalize to more complex applications such as semantic segmentation [Pándy et al., 2022, Agostinelli et al., 2022]. With the introduction of new methods utilizing frozen VLMs for detection (e.g. “F-VLM: Open-Vocabulary Object Detection upon Frozen Vision and Language Models”, ICLR 2023) and segmentation (e.g., “Open-vocabulary Semantic Segmentation with Frozen Vision-Language Models”, BMVC2022 and “Convolutions Die Hard: Open-Vocabulary Segmentation with Single Frozen Convolutional CLIP”, arXiv 2023), LOVM methods would most likely transfer well and classification performance would be indicative of performance in downstream tasks.
>
> As LOVM methods continue to advance, we foresee the development of future LOVM benchmarks that build upon the one introduced here, incorporating tasks like segmentation and detection. Such future work might also leverage future open-source repositories combining many model variations (e.g., open_clip), broadening their scope to multimodal detection and segmentation.
>
>
> **2. Fine-Tuning VLMs with Large-Scale Datasets:**
>
> We agree with your perspective regarding CVML researchers working with benchmark datasets. However, our primary focus is application-driven practitioners who struggle to collect large-scale datasets akin to ImageNet, and therefore use these VLMs in the zero-shot setting. LOVM could be a valuable tool for those with limited access to such datasets or resources to run exhaustive evaluations.
>
> Our envisioned use-case illustrates this in Fig. 2 of the manuscript. If LOVM is successful, users can input a task description into a search bar (e.g., on the OpenAI/Transformers website) and receive the optimal model selection and an estimated performance for their specific task.
>
> *We have re-drafted L48-L64 of the introduction to clarify this important aspect of our work.*
>
> Previous transferability work has seen correlations between zero-shot and few-shot (linear probing) of pre-trained VLMs and their downstream performance [Wong et. al, arXiv 2022], but this process would introduce additional noise to LOVM performance prediction & is an interesting topic for future work - please see the first paragraph of the limitations section (Sec. D, Appendix).
>
>
>
> **3. Small Differences in Ground Truth Accuracy Rates:**
>
> Your observation is astute. This is caused by the inclusion of very similar VLMs, such as different checkpoints of the same model trained on the same dataset but with different augmentation. However, while these differences would affect k_tau, our other two metrics, specifically R_5 and L_1, would not be as affected. Additionally, by averaging the ranking performance across 23 datasets, the noise introduced by small variations is substantially reduced.
>
> We are grateful for your thoughtful critique and will incorporate your feedback to improve our manuscript.
>
> Best regards,
>
> The Authors

---

> > ### Author Response · Authors · 2023-08-12
> > **Additional Experiments - effect of noise on evaluation and metrics**
> >
> > Dear R-nEyK,
> >
> > To address your concerns, we have run additional experiments. Specifically:
> >
> > **3. Small Differences in Ground Truth Accuracy Rates:**
> >
> > We added random Gaussian noise to the GT evaluation (N[0, 0.01] and N[0, 0.1]) 100 times, ran the LOVM baselines, and computed the mean and STD for predicting top-1 accuracy ranking R5 and k_tau on the proposed LOVM benchmark:
> >
> > |        | sigma=0.1 R5 (↑) | sigma=0.1 tau (↑) | \| |sigma=0.01 R5 (↑) | sigma=0.01 tau (↑) |
> > | ------         |     ----------------     | ----------------- | --- | ----------------- | ------------------ |
> > | C             | 0.226 ± 5.5e-17   | 0.058 ± 1.0e-16    | \| | 0.226 ± 5.5e-17    | 0.058 ± 1.0e-16     |
> > | G             | 0.252 ± 2.8e-16   | -0.014 ± 1.0e-17   |\|  | 0.252 ± 2.8e-16    | 0.014 ± 1.0e-17     |
> > | G+C        | 0.252 ± 2.8e-16   | -0.014 ± 1.0e-17   | \| | 0.252 ± 2.8e-16    | 0.014 ± 1.0e-17     |
> > | INB+C     | 0.454 ± 0.008     | 0.214 ± 0.005      | \| | 0.460 ± 0.0009     | 0.222 ± 0.003      |
> > | INB+G     | 0.454 ± 0.016     | 0.118 ± 0.03       |\| | 0.461 ± 0.003      | 0.096 ± 6.9e-17     |
> > | INB+G+C | 0.454 ± 0.008     | 0.214 ± 0.005      |\| | 0.460 ± 0.0009     | 0.222 ± 0.003      |
> >
> > We hope these experiments demonstrate that small errors in evaluation will not drastically affect the overall evaluated k_tau and R_5. The small STD is due to the fact that we are averaging the ranking over 23 datasets, and this only has an effect if the differences in the ranking of models are in the top-5.
> >
> > While the std is relitively small, we can clearly see the mean being affected when we introduce a large ammount of noise (N[0, 0.1]) - e.g., 0.461->0.454 mean for INB+G+C. As the top-1 accuracy is in the range of [0, 1], this noise is very substantial and effectively corrupts the correlation between the models and the GT+noise performance - further indicating that LOVM methods learn non-trivial model rankings.
> >
> > Best regards,
> >
> > The Authors

---

> > > ### Comment · Reviewer_nEyK · 2023-08-22
> > > **Thanks for the authors' feedback**
> > >
> > > The authors' response effectively resolves my concerns. As a result, I am inclined to increase my score to 7.

---

> > > > ### Author Response · Authors · 2023-08-24
> > > >
> > > > Dear R-nEyK,
> > > >
> > > > We want to express our heartfelt gratitude for the constructive feedback you've provided and for re-evaluating our paper. Your insights have been invaluable in strengthening our work, and we are grateful for the time and thought you have invested in understanding LOVM. We also want to thank you for your willingness to reconsider and increase our score. We genuinely appreciate your support.
> > > >
> > > > Best regards,
> > > >
> > > > The Authors

---

### Official Review · Reviewer_i9wv · 2023-07-23
**Novel benchmark design and methodological contributions**

**Rating:** 7
**Confidence:** 4
**Correctness:** Yes.
**Clarity:** Yes.

**Strengths:**

1. Overall, the paper is well structured and easy to follow.
2. Extensive models and datasets are included in the benchmark.
3. The evaluation metrics and protocols are well designed.

**Additional Feedback:**

See Opportunities For Improvement.

**Documentation:**

Yes.

**Ethics:**

No.

**Limitations:**

See Opportunities For Improvement.

**Opportunities For Improvement:**

1. The paper claims that the proposed benchmark offers an efficient solution for model selection, but there is no specific quantification provided regarding time reduction or computation reduction compared to traditional evaluation methods.
2. The focus of this work on vision tasks is likely due to the increasing popularity of multi-modal vision-language models and their exceptional performance on vision-related downstream applications. Could the proposed method be adapted to natural language tasks?
3. It's mentioned that this work used readily-available large language models to generate 'text datasets' for a given vision task. Is there any uncertainty inside the generated 'text datasets'? Will that affect the final evaluation results of VLMs?

**Relation To Prior Work:**

Yes.

**Summary And Contributions:**

This paper addresses the challenge of evaluating numerous available VLMs by proposing Language-Only Vision Model Selection (LOVM) for assessing VLMs' zero-shot performance on downstream applications without access to the actual downstream task dataset. The study introduces an extensive LOVM benchmark, where methods are expected to rank the VLMs and predict their zero-shot performance using only a text description of the desired downstream application, along with corresponding evaluation metrics and protocols. Additionally, the methodological contributions demonstrate the superiority of text-based methods over ImageNet benchmarking. Furthermore, the paper offers valuable insights into VLM behavior, shedding light on why ResNet-based models outperform others on datasets with low visual diversity.

---

> ### Author Response · Authors · 2023-08-08
>
> Dear R-i9wv,
>
> Thank you for your thoughtful review and feedback on our paper. We truly value the time and expertise you've devoted to this process. Below, we address each of your points.
>
> **1. Quantification of Efficiency of the Proposed Benchmark:**
>
> We are grateful to R-i9wv for their thoughtful observations and for bringing our attention to the important issue of our baseline's efficiency. **In the context of LOVM, it's crucial to recognize that the predominant advantage lies not in computational savings but in the alleviation of the need for assembling and annotating datasets to directly assess model performance.** Assembling and labeling a novel task-specific validation dataset, such as a classification set with 50,000 images, is a process that is both expensive and time-consuming. For example, using Amazon Mechanical Turk to annotate 50,000 images costs approximately ~7k USD [1] and could span several weeks to months. The cost of annotating specialized datasets, e.g., medical, is even more time-consuming and expensive. This also assumes that one incurs no cost in collecting the images.
>
> [1] https://calculator.aws/#/estimate
>
> In stark contrast, utilizing LOVM to evaluate a dataset incurs a cost of less than 1 USD and can be performed in a matter of minutes (depending mostly on the token rate limit in text generation), representing substantial savings in both time and financial resources. While it is true that the text encoders in CLIP models possess fewer parameters than the image encoders, rendering LOVM less computationally demanding per evaluation, this is not the primary source of the savings. The primary value of LOVM stems from its ability to significantly reduce the logistical and financial burdens associated with dataset collection and annotation. By focusing on this aspect, LOVM delivers immense practical utility and relevance.
>
> The total compute and run time on a traditional classification task is dependent on the number of images in the dataset. When comparing the total time it took to evaluate all the VLMs on all of the datasets to the time it took for us to run all the LOVM experiments, we found that (*our*) LOVM baselines were ~7x faster than image-based evaluation on the same machine. Future LOVM methods may be less/more efficient than our own.
>
> *We have added this information in the “Baseline Details” section of the appendix.*
>
>
> **2. Adaptability to Natural Language Tasks:**
>
> Your observation is insightful. The proposed LOVM method, while initially conceived for vision tasks, has the potential to be adapted to natural language tasks, especially when it comes to multi-modal LLMs. Exploring this direction could be a fascinating avenue for future work, and we thank you for highlighting it.
>
>
> **3. Uncertainty in Generated 'Text Datasets':**
>
> Indeed, some uncertainty exists in the generated 'text datasets'. We did perform a qualitative comparison between datasets generated using chatgpt-3.5 and those from DaVinci (which we originally utilized). Interestingly, there wasn't a substantial difference in the image descriptions, leading us to select the more cost-effective alternative (chatgpt-3.5) for our study.
>
> We thank R-i9wv for their valuable comments and insights.
>
> Best regards,
>
> The Authors

---

> > ### Author Response · Authors · 2023-08-12
> > **Additional Experiments - effect of LLM on LOVM baselines**
> >
> > Dear R-i9wv,
> >
> > We have re-run our baselines with a different LLM (gpt-3.5-turbo-16k vs gpt-3.5-turbo-0301) and have gotten the following results:
> >
> > **3. Uncertainty in Generated 'Text Datasets':**
> >
> > | used    |                                    | mean per-class recall |                                           | \|   |                             | top-1 accuracy |                             |
> > | ------- | ----------------------------- | --------------------------- | ---------------------- | --- | -------------------- | -------------------- | -------------------- |
> > | scores  | $R_5 (\uparrow)$             | $\tau(\uparrow)$      | $L_1(\downarrow)$     | \| | $R_5(\uparrow)$ | $\tau(\uparrow)$ | $L_1(\downarrow)$ |
> > | INB                   | 0.504                        | 0.186                 | 0.228                 |\|  | 0.452          | 0.177           | 0.220           |
> > | C                      | 0.365                        | 0.072                 | 0.144                 |\|  | 0.357          | 0.043           | 0.145           |
> > | G                      | 0.252                        | 0.014                 | 0.133                 |\|  | 0.243          | 0.029           | 0.135           |
> > | G+C                 | 0.365                        | 0.072                 | 0.133                 | \| | 0.357          | 0.043           | 0.129           |
> > | INB+C              | 0.504                        | 0.223                 | 0.144                 |\|  | 0.461          | 0.200           | 0.145           |
> > | INB+G              | 0.522                        | 0.191                 | 0.133                 | \| | 0.461          | 0.212           | 0.135           |
> > | INB+G+C.        | 0.522                        | 0.191                 | 0.133                 | \|  | 0.470          | 0.078           | 0.129           |
> >
> >
> > As can be seen, using a different LLM is a design choice that affects performance; however, the overall conclusions remain the same. As LLMs continue to improve, and perhaps with better prompting strategies, the quality of the "text datasets" will increase, making LOVM methods more accurate.
> >
> > Best regards,
> >
> > The Authors

---

> > > ### Author Response · Authors · 2023-08-24
> > >
> > > Dear R-i9wv,
> > >
> > > We're grateful for your detailed review and the positive evaluation of our paper, "LOVM: Language-Only Vision Model Selection."
> > >
> > > In response to your feedback, we conducted additional experiments and addressed your concerns in the previous comments. We sincerely hope that these further insights and clarifications meet your expectations, and we kindly ask if these additions might lead you to consider an increased score for our work.
> > >
> > > With just five left for the reviewer-author discussion period, we wanted to ensure you have everything you need for your evaluation. Please don't hesitate to reach out if you have any further questions or need additional information.
> > >
> > > Thank you again for your time and consideration.
> > >
> > > Best regards,
> > >
> > > The Authors

---

### Official Review · Reviewer_U8rx · 2023-07-23
**LOVM: Language-Only Vision Model Selection**

**Rating:** 6
**Confidence:** 3
**Correctness:** Yes

**Strengths:**

The paper tackles a valuable and compelling topic.
The concept of using language-only VLM selection and performance prediction is novel and intriguing.
The paper is well-structured, and the authors provide a clear and detailed description, enhancing comprehension.

**Additional Feedback:**

None

**Clarity:**

The paper is generally well-organized and clearly presented.



**Documentation:**

Yes

**Limitations:**

The authors have thoroughly discussed the limitations and potential negative social impacts of their work.

**Opportunities For Improvement:**

The authors need to clarify the motivation of using text to roughly predict the accuracy of a relevant task on an image dataset. A detailed explanation in the introduction section, possibly supplemented by supportive experimental results, would significantly improve understanding.
The absence of accessible code and datasets limits the ability to assess and verify the proposed methods and benchmarks.
The format of references requires attention, particularly for conference papers. For instance, references [5], [8], and [9] need revision.

**Relation To Prior Work:**

The authors have clearly distinguished their dataset from previously existing ones.



**Summary And Contributions:**

The authors of this paper propose an innovative task titled LOVM (Language-Only Vision Model Selection). This task necessitates performing both model selection and performance prediction using merely the textual description of the intended downstream application. In addition, the authors present a comprehensive LOVM benchmark composed of ground-truth evaluations of 35 pre-trained Vision Language Models (VLMs) across 23 datasets. The requirement is to rank these pre-trained VLMs and predict their zero-shot performance. The rationality and effectiveness of this method are demonstrated through experiments.

The paper has two main contributions:

The introduction of a novel task, LOVM: Language-Only Vision Model Selection, where selection and performance prediction is achieved solely through text description.
The establishment of a substantial benchmark comprising 35 pre-trained VLMs and 23 datasets.

---

> ### Author Response · Authors · 2023-08-08
>
> Dear R-U8rx,
>
> We sincerely appreciate your insightful comments and the time you invested in reviewing our manuscript. We're encouraged by your acknowledgment of the paper's value and the novelty of the LOVM task. Below, we address each of your points of concern.
>
> **1. Clarification on the Motivation of Using Text to Predict Accuracy on an Image Dataset:**
>
> If the intention of R-U8rx was in regard to the motivation of LOVM as a whole, please see the response to all reviewers.
>
> If R-U8rx was inquiring regarding the transferability phenomenon, we updated the “text as a proxy for images” paragraph in the related works section to clarify that this is where we discuss the cross-modal transferability phenomenon, which is the foundation of why text input can be used as a proxy for images in VLMs.
>
> L301 was changed: “Text as a Proxy For Images.” -> “The Cross-Modal Transferability Phenomenon: Text as a Proxy For Images.”
>
> There, we discuss: “...Subsequently, Zhang et al. [2023] has found that this gap can be approximated as an orthogonal constant between true pairs of image and text and is, therefore, parallel to the decision boundaries for a given modality. This suggests that cross-modality transferability - using one modality as input to the other’s classifier - is possible for these contrastively pre-trained VLMs. Several studies have demonstrated the utility of the cross-modality transferability phenomenon in different tasks….”
>
>
> **2. Absence of Accessible Code and Datasets:**
>
> The code and dataset are available at https://github.com/orrzohar/LOVM. There you will find all the code required for reproducing our work, as well as the LOVM benchmark. We sincerely apologize for any confusion caused and have updated the manuscript to highlight our Github link.
>
>
> **3. Formatting of References:**
>
> Thank you for pointing this out. We have thoroughly reviewed the entire references to make sure that they are uniform.
>
>
> We hope these revisions will align our work more closely with the conference's high standards, and we thank you for your valuable feedback.
>
> Best regards,
>
> The Authors

---

> > ### Author Response · Authors · 2023-08-24
> >
> > Dear R-U8rx,
> >
> > Thank you for reviewing our paper "LOVM: Language-Only Vision Model Selection." We appreciate your time and effort in reviewing our work and ensuring it meets the high standards expected in the NeurIPS community.
> >
> > As the end of the reviewer-author discussion period is approaching with just five days left, we wanted to reach out and see if you have any additional questions or need further clarification. We're here and happy to respond.
> >
> > We hope that the points addressed above are sufficient to increase your confidence in our work.
> >
> > Thank you for your time and consideration.
> >
> > Best Regards,
> >
> > The Authors

---

### Author Response · Authors · 2023-08-08
**Response to All Reviewers**

We thank the reviewers for their valuable feedback and are pleased to see that they found the LOVM task to be valuable, compelling and practically useful (R-U8rx, R-nEyK, R-BH74), that the LOVM benchmark is comprehensive (R-U8rx, R-i9wv, R-nEyK, R-BH74), that the code was open-source and well documented (R-nEyK, R-BH74), and that the paper was well-structured and presented (R-U8rx, R-i9wv, R-BH74).

Given the reviewers' feedback, we have revised the manuscript to improve the clarity of the motivation for and possible applications of the LOVM task - and have revised the L28-L34 & L48-L61 of the Introduction. Briefly, as the performance of Vision-Language Models (VLMs) in the zero-shot setting continues to improve, one can adapt these models to a downstream task via text prompting. ML practitioners are therefore able to develop applications without needing to train a model simply by changing the text prompts used by the VLM. However, with the number of pre-trained VLMs growing exponentially, it is becoming increasingly difficult to select the optimal VLM for a downstream task. Currently, to deploy/use a VLM for a downstream task one has to: (1) collect an evaluation dataset, (2) label that dataset, and (3) exhaustively evaluate all available VLMs to select the optimal model. **Of the three, (1) and/or (2) (data collection, curation, and labeling) tend to be the most time-consuming and expensive.**

This is where LOVM comes in. **With LOVM, ML application researchers can (by providing a text description of their target application) select the optimal VLM as well as get an estimate of its performance – all without ever needing to collect and annotate an image-based evaluation dataset.** They then could directly proceed to that target application and deploy/use the model - quickly proceeding to the next stage of their research and/or application. An example of such a downstream/future “LOVM application” is the addition of a 'VLM Search bar' in the Transformers website where a user would describe a target application and then would get a list of promising VLMs, ordered by their predicted performance to select from.

We are deeply grateful for your thoughtful feedback and constructive criticisms of our manuscript and the proposed LOVM benchmark. Your insights are instrumental in ensuring our work aligns with the high standards and expectations of the NeurIPS community.

Best regards,

The Authors

---

> ### Author Response · Authors · 2023-08-12
> **Additional Experiments**
>
> We have conducted additional experiments (requested by R-i9wv and R-nEyK) and the results are posted as comments to their reviews/discussions.
>
> Best regards,
>
> The Authors

---

### Decision · Program_Chairs · 2023-09-22

**Decision:**

Accept (Poster)

**Comment:**

This paper introduces a novel and practically useful problem that only text descriptions of the target task are used to select vision models. To this end, a benchmark is introduced to efficiently evaluate VLMs' zero-shot performance on downstream applications without access to the downstream task dataset.
We had four reviewers and their ratings were all positive: 6, 7, 7, 7. All reviewers agree that the paper makes some substantial contributions to the community. Based on the reviewers' comments and the author's response, I recommend acceptance.